



# Local comparisons of tropospheric ozone: Vertical soundings at two neighbouring stations in Southern Bavaria

Thomas Trickl[1], Martin Adelwart[2], Dina Khordakova[3], Ludwig Ries[4], Christian Rolf[3], Michael Sprenger[5], Wolfgang Steinbrecht[2] and Hannes Vogelmann[1]

[1]Karlsruher Institut für Technologie, Institut für Meteorologie und Klimaforschung (IMK-IFU), Kreuzeck-bahnstr. 19, D-82467 Garmisch-Partenkirchen, Germany

[2]Deutscher Wetterdienst, Meteorologisches Observatorium, Albin-Schwaiger-Weg 10, 82383 Hohenpeißenberg, Germany

[3]Forschungszentrum Jülich, IEK-7, Wilhelm-Johnen-Straße, 52425 Jülich, Germany

[4]Umweltbundesamt II 4.5, Plattform Zugspitze, GAW-Globalobservatorium Zugspitze-Hohenpeißenberg, Schneefernerhaus, 82475 Zugspitze, Germany

[5]Eidgenössische Technische Hochschule (ETH) Zürich, Institut für Atmosphäre und Klima, Universitätstraße 16, 8092 Zürich, Switzerland

*Correspondence to:* Dr. Thomas Trickl, thomas@trickl.de, Thomas-Knorr-Str. 47, D-82467 Garmisch-Partenkirchen, Germany; tel. +49-8821-50283; Dr. Hannes Vogelmann, hannes.vogelmann@kit.edu, Karlsruher Institut für Technologie, IMK-IFU, Kreuzeckbahnstr. 19, D-82467 Garmisch-Partenkirchen, Germany; tel: +49-8821-258

**Abstract.** In this study ozone profiles of the differential-absorption lidar at Garmisch-Partenkirchen are compared with those of ozone sondes of the Forschungszentrum Jülich and of the Meteorological Observatory Hohenpeißenberg (German Weather Service). The lidar measurements are quality assured by the highly accurate *in-situ* measurements at nearby the Wank (1780 m a.s.l.) and Zugspitze (2962 m a.s.l.) summits and at the Global Atmosphere Watch station Schneefernerhaus (2670 m a.s.l.). The lidar results agree almost perfectly with those of the monitoring stations. Side-by-side sounding of the lidar and electrochemical (ECC) sonde measurements by a team of the Forschungszentrum Jülich shows just small positive offsets (≤ 3.4 ppb), almost constant within the troposphere. We conclude that the recently published uncertainties of the lidar in the final configuration since 2012 are realistic and rather small for low to moderate ozone. Comparisons with the Hohenpeißenberg routine Brewer-Mast sonde measurements are more demanding because of the distance of 38 km between both sites. These comparisons cover the three years September 2000 to August 2001, 2009 and 2018. A slight negative average offset (−3.64 ppb ± 7.5 ppb (full error)) of the sondes with respect to the lidar is found. Most sonde measurements could be improved in the troposphere by recalibration with the station data. This would not only remove the average offset, but also greatly reduce the variability of the individual offsets. The comparison for 2009 suggests a careful partial re-evaluation of the lidar measurements between 2007 and 2011 for altitudes above 6 km where an occasional negative bias occurred.

*Key words:* Tropospheric ozone, ozone sonde, lidar, differential absorption

## 1. Introduction

The development of tropospheric ozone has been studied over more than a century (e.g., Gaudel et al., 2018; Tarasick et al., 2019). For many decades, balloon-borne ozone sondes have been a primary work horse of ozone profiling. Their measurement principle is based on the oxidation of iodide ($I^-$) to iodine ($I_2$) by ozone in a wet-chemical potassium iodide (KI) cell. Between cathode and anode of the wet-chemical cell, the oxidation reaction drives an electrical current which can be measured (two electrons per ozone molecule). Recently, nearly all



stations have used the so-called ECC (electro-chemical-cell) sonde type (Komhyr 1969; 1995), featuring two
cells with different potassium iodide concentrations (anode and cathode cell). Only the Hohenpeißenberg station,
discussed here, still uses the older-type Brewer-Mast sondes (Brewer and Milford, 1960), which uses one cell
only (with a platinum cathode and a silver anode), and a less efficient pump design (Steinbrecht et al., 1998).
Ozone sondes have been characterized in numerous studies, both in flight (e.g., Attmannspacher and Dütsch,
1981; De Muer and Malcorps, 1984; Beekmann et al., 1994; Kerr et al., 1994; Jeannet et al., 2007; in recent
years: Gaudel et al., 2015; Van Malderen et al., 2016; Deshler et al., 2017; Tarasick et al., 2021; Ancellet et al.,
2022; Stauffer et al., 2022), and in a laboratory simulation chamber (Smit et al., 2007, 2014, 2021). Generally,
the accuracy of individual ECC soundings for ozone in the mid-latitude troposphere is about 5 to 10% (Logan et
al., 2012; Smit et al., 2014; Tarasick et al., 2016, 2019). Following rigorous best practices, 5% accuracy can be
achieved (Vömel et al., 2020; Smit et al., 2021; Tarasick et al., 2021). For Brewer-Mast soundings, the accuracy
is slightly lower in the troposphere, about 10 to 15% (Smit et al. 2014; Tarasick et al., 2016, 2019).
The ozone soundings at the Meteorological Observatory Hohenpeißenberg (MOHp) of the German Weather
Service (Deutscher Wetterdienst, DWD) in Southern Bavaria have been routinely carried out since November
1966, yielding one of the longest ozone-sonde time series. Brewer-Mast ozone sonde data tend to have a low
bias above about 25 km altitude (Steinbrecht et al., 1998). In the troposphere, compared to ECC soundings,
Tarasick et al. (2002, 2016) find a negative bias of about 20 % for ozone from Canadian Brewer-Mast soundings
prior to 1980. European Brewer-Mast stations, however, have generally used a much more extensive preparation
procedure for their sondes (Claude et al. 1987), and no significant tropospheric bias has been reported for their
routine Brewer-Mast soundings (de Backer et al. 1998; Stübi et al. 2008; Logan et al., 2012), as well as in
chamber experiments (Smit et al., 2014).
Routine measurements with ozone sondes yield time series free of a fair-weather bias. However, the balloon
ascents take place at intervals of several days. Ozone profiles at short intervals can be provided by lidar
sounding, but are limited to clear atmospheric conditions. Lidar measurements can generate altitude-time curtain
plots and, thus, give much better insight into the impact of atmospheric transport (e.g., Browell et al., 1987;
Ancellet et al., 1991; Langford et al., 1996).
At IFU (Fraunhofer-Institut für Atmosphärische Umweltforschung; now: Karlsruher Institut für Technologie,
IMK-IFU) in Garmisch-Partenkirchen (Germany), a differential-absorption lidar (DIAL) with a particularly wide
operating range from next to the ground up to the upper troposphere was completed in 1990 in the framework of
the TESLAS (Tropospheric Environmental Studies by Laser Sounding) subproject of EUROTRAC (TESLAS,
1997; EUROTRAC, 1997, Kempfer et al., 1994). Subsequently, the system was applied for a full year (1991)
within the TOR (Tropospheric Ozone Research; Kley et al., 1997) subproject of EUROTRAC (Carnuth et al.,
2002). The operating range of this system was extended upwards to roughly 15 km in 1994 by introducing three-
wavelength operation (Eisele et al., 1999).
Until 2003 the system was used for individual research projects. Between 2007 and 2018 routine measurements
took place, parallel to lidar measurements of water vapour (Vogelmann and Trickl, 2008) and aerosol. The
complementary information from these instruments has made possible a large number of investigations related to
atmospheric transport. The IFU ozone DIAL was recently fully described by Trickl et al. (2020a).
The distance between MOHp and IFU is just 38 km which offers a good chance for comparisons. Due to its
design, the IFU ozone DIAL features particularly low uncertainties. However, such a comparison must be made
with care since the atmospheric variability is high on a rather small scale (Vogelmann et al., 2011; 2015), mostly
caused by the advection of air masses from rather different source region and altitudes, with different



concentrations (e.g., Stohl and Trickl, 1999; Trickl et al., 2003; Trickl et al., 2011). The variability of the vertical
distribution of ozone measurements rarely yields very strong concentration changes, but the concentration
changes are extreme for water vapour (Vogelmann et al., 2011; 2015). Our lidar measurements of water vapour
exhibit a concentration span of more than two decades, with minima of the relative humidity (RH) clearly below
1 % in layers descending from the stratosphere (Trickl et al., 2014; 2015; 2016; Klanner et al., 2021).
Comparisons between the MOHp sonde and the IFU lidar were made in the second half of the 1990s and in
2001, after the first upgrading of the lidar A few of these comparisons in 1996 and 1997 were published by
Eisele et al. (1997; 1999). For the six cases with sufficient air-mass matching a principal agreement in the middle
and upper troposphere to within 5 ppb prevailed with occasional departures of the order of 10 ppb.
Hints on ozone differences between the Zugspitze (2962 m a.s.l..) *in-situ* data and the MOHp values (H. E.
Scheel, personal communication around 2010) for 3 km a.s.l. have led to a revived interest in a thorough
comparison. There have been speculations about an influence of a different air composition outside the
mountains at low altitudes up to a few kilometres.
In this paper we first characterize the lidar performance by side-by-side ascents of ozone-sondes by a team of the
Forschungszentrum Jülich (FZJ). Then, we give a statistical assessment for the measurements at IFU and MOHp
for 2018. For this year we achieved the best coverage of by DIAL measurements. This allows us to make an air-
mass related data selection to improve the comparison. After the shutdown of the IFU stations in 2012,
comparisons have been made exclusively with the Global Atmosphere Watch (GAW) routine *in-situ*
measurements at the Schneefernerhaus high-altitude station (UFS, 2671 m a.s.l.). UFS is located just below the
Zugspitze summit. Finally, we also compare lidar and MOHp sonde for two earlier development phases of the
lidar, for which ozone reference data at the local summit stations Wank (1780 m a.s.l.) and Zugspitze exist.
**2. Methods**
**2.1 Brewer-Mast sonde system at Hohenpeißenberg**
MOHp is located on an isolated mountain outside the Alps, 38 km to the north of IMK-IFU and 50 km to the
south-west of Munich (975 m a.s.l., 47.80 N, 11.00 E). Brewer-Mast ozone sondes have been launched on a
regular basis since November 1966. The sondes undergo a rigorous preparation procedure (Claude et al. 1987),
which has remained essentially unchanged since the early 1970s. From 1995 to 2005, Vaisala RS80 radiosondes
and a Vaisala PC-CORA ground station have been used in combination with the ozone sondes. This was
changed to Vaisala RS92 radiosondes and DigiCora III MW31 ground equipment in 2005, to MW41 ground
station in 2018, and to Vaisala RS41 radiosondes in 2019. The standard processing does not subtract a
background current, but ozone sondes with non-negligible background current on the ground (> 2.5 ppb ozone)
are not flown. Pump temperature is assumed to be constant at 300 K, which compensates to some degree for a
too weak pump correction in the stratosphere (Steinbrecht et al. 1998). A time-lag correction is not applied, but
this is not critical outside regions with steep ozone gradients. Each ozone profile is adjusted by multiplication
with an altitude-independent correction factor, so that the total ozone column estimated from the sounding
(including an extrapolation above approximately 30 km) matches the more accurate total ozone measurement
from on-site Dobson or Brewer spectrometers, or from satellite instruments. This so-called "Dobson correction"
generally improves that accuracy of the ozone sounding data in the stratosphere, but may introduce a small bias
in the tropospheric data of some soundings (e.g., Stübi et al., 2008; Logan et al. 2012).
The MOHp ozone sonde and radiosonde data are stored in the data base of the Network for the Detection of
Atmospheric composition change (NDACC), from where they were imported for the study presented here.



**2.2 ECC sonde system of the Forschungszentrum Jülich (FZJ)**
A mobile ballon-borne sonde system of FZJ was operated at IMK-IFU (at 730 m a.s.l.), in close vicinity to the
ozone DIAL, during the FIRMOS measurement campaign (Klanner et al., 2020; Palchetti et al., 2021; Di Natale,
2021). Several balloons with cryogenic frostpoint hygrometers (CFH; Vömel et al., 2007; 2016), standard
Vaisala RS-41-SGP radiosondes (Vaisala et al., 2019), En-Sci ECC ozone sondes (Komhyr et al., 1995; Smit et
al., 2007) and COBALD backscatter sondes (Brabec, 2011) were launched. The data were transmitted to a
ground station installed for this campaign at the Zugspitze summit. The combined balloon payload is well tested
and regularly also used by the GCOS Reference Upper Air Network (GRUAN) (e.g., Dirksen et al., 2014).
We followed the standard operating procedures (SOP) of Smit et al. (2014) for the sonde preparation using a
solution composition of 1 % and 1/10 (one-tenth) buffer for best results with sondes from the manufacturer En-
Sci (Thompson et al., 2019).
For the analysis of the ECC data, the methods described by Vömel et al. (2020) are used, i.e., time lag correction
and background current correction. The overall uncertainty of the ozone measurements of the ECC sondes is 5%.
Due to the obstruction of the line of sight between between launch site at IMK-IFU and the ground station at the
summit by the Waxenstein mountain allowed data recording only from approximately 1500 m altitude upwards.
Therefore, we used the estimated ECC background current from the sonde preparation one day before a flight as
starting value for the background correction instead of the actual measured profile from ground up to 1500 m.
This results in an additional uncertainty in the lower part of the profile (2 to 3 km a.s.l.).
**2.4 IFU ozone DIAL system**
The ozone DIAL of IMK-IFU (Garmisch-Partenkirchen), located at 47.477 N, 11.064 E, and 740 m a.s.l., has
been developed and optimized since 1988 (Kempfer et al., 1994; Trickl et al., 2020a). It is based on a krypton
fluoride excimer laser, operated at 400 mJ per pulse (40 W) of narrowband radiation at 248.5 nm, two
Newtonian receiving telescopes (diameter of the primary mirrors: 0.13 m and 0.5 m) and 1.1-m grating
spectrographs for wavelength separation. Efficient stimulated Raman shifting in hydrogen and deuterium yields
emission at the three operating wavelengths 277.2 nm, 291.8 nm and 313.2 nm. The shorter-wave spectral
components are absorbed by ozone ("on" wavelengths), that at 313.2 nm ("off" or reference wavelength) is
almost outside the absorption region of $O_3$. The laser system is operated with a repetition rate of 99 Hz which
allows a short data-acquisition time of just 41 s.
The shortwave 277.2-nm emission yields particularly accurate measurements, but the strong extinction of this
radiation by ozone limits the range to about 8 km. The performance in the two 277.2-nm channels is robust with
respect to minor misalignment, with uncertainties of about 2 to 3 ppb. This is not the case for 291.8 nm where
the optical alignment must be controlled with care because of less tight focussing into the entrance slit of the far-
field spectrograph. In addition, the 291.2-nm backscatter signal is three times noisier than that for 277.2-nm. The
noise of the 313.2-nm signal becomes important at large distances. As a consequence, the uncertainty of the
ozone mixing ratio can be become rather high in the upper troposphere and the tropopause region, in particular in
summer due to the stronger loss of signal caused by the higher levels of ozone. Sometimes the uncertainty just
below the tropopause can even exceed 10 ppb.
The DIAL data processing is made for different wavelength combinations (Eisele and Trickl, 2005). In this way,
an internal quality control can be achieved. The optical alignment is optimized immediately after detecting an
ozone mismatch in the first quicklook data evaluation. Just the laser beam overlap of the different wavelength
components (Trickl et al., 2020a) and the beam pointing must be optimized.



The calibration of the lidar measurements has been based from the very beginning (1991) on the accurate
temperature-dependent ozone absorption cross sections of the University of Reims (Daumont et al., 1992;
Malicet et al., 1995). These cross sections were verified for four wavelengths below 300 nm by Viallon et al.
(2015) to within ±0.06 %. In the presence of aerosol an aerosol correction is made with the algorithms of Eisele
and Trickl (2005). This correction is rather robust for the wavelength pair 277 nm - 292 nm because of the strong
absorption at the short "on" wavelength and the moderate wavelength difference (Völger et al., 1996).
Meteorological data for calculating density and temperature profiles are taken from the Munich radiosonde
(station 10868). The retrieved 313-nm aerosol backscatter coefficients have been routinely stored in the data
base of the European Aerosol Lidar Network (EARLINET) since 2007.
After repeated system upgrading the final performance of the lidar was reached in late 2012. In the absence of
aerosol the far-field ozone could be evaluated with high reliability from the 291.9-nm signal alone, after
precisely modelling the air number density from radiosonde data. In this way the daytime noise induced by the
division by the 313-nm data in summer could frequently be avoided.
During the final decade of the lidar operation a fitting procedure was applied in noisy situations in the upper
troposphere (i.e., under high-ozone conditions in summer). This procedure reduces unrealistic curvature of ozone
structures caused by enhanced data smoothing, and, thus, abrupt concentration changes (in particular at the
tropopause) visible in the raw data are reproduced in the mixing ratio. We prepared an extension of the data-
acquisition time from 41 s to about five minutes in order to improve the signal-to-noise ratio. However, the lidar
operation ended before the start of this option.
From 1991 to 2003 the DIAL was operated for focussed research projects. Routine measurements took place
from 2007 to 2018, until 2015 parallel to measurements with a water-vapour DIAL (Trickl et al., 2014, 2015,
2016, 2020b). In 2012 the highest data quality was finally reached, which included significant improvements for
the near-field telescope (Trickl et al., 2020a). Thus, the conditions for a meaningful system validation were
obtained. The operation was discontinued in February 2019, after the retirement of the first author of this paper.
**2.5 *In-situ* measurements**
Quality-assured ozone measurements at the Wank (1780 m a.s.l., 7.0 km to the north-east of IMK-IFU, 47.511º
N, 11.141º E) and Zugspitze (2962 m a.s.l., 8.4 km to the south-west of IMK-IFU, 47.421º N, 10.986º E) took
place from 1978 to 2012. Since the 1990s two or three TE 49 analysers (Thermo Environmental Instruments,
USA) were operated simultaneously at each station. These instruments are based on ultraviolet (UV) absorption
at 253.65 nm. Several comparisons using transfer standards (O$_3$ calibrators TE 49 PS) were made with the World
Meteorological Organization (WMO) Global Atmosphere Watch (GAW) reference instrument kept at the
WMO/GAW calibration centre operated by EMPA, Switzerland (Klausen et al., 2003). The most recent
comparison was conducted in June 2006 and confirmed that the Zugspitze O$_3$ data are on the GAW scale.
Apart from the two mountain stations measurements were performed also at IFU at about 740 m a.s.l. (47.477º
N, 11.064º E). This laboratory was adjacent to that of the ozone DIAL.
At UFS (0.70 km to the south-east of Zugspitze, 47.417º, 10.980º E) ozone has been continuously measured
since 2002 by a team of the German Federal Environment Agency (Umweltbundesamt, UBA) using TEI 49i
instruments (Thermo Electron Corporation). The gas inlet is at 2671 m a.s.l. As ozone standard for weekly and
monthly calibration a TEI 49C-PS instrument was applied that was calibrated against the ozone standard of UBA
(UBA SRP#29) on an annual basis. UBA operates the German standard for Ozone. It was adjusted via BIPM
(Bureau International des Poids et Mesures) in Paris to the NIST ozone reference standard of the WMO/GAW





measurement programme. The measurements were supported by a second instrument (Horiba APOA-370). The
instrumentation is fully adequate for Global Atmosphere Watch monitoring. GAW system and performance
audits at the station for surface ozone took place in 2001, 2006 and 2011.
The uncertainty of the *in-situ* ozone measurements is ±0.5 ppb with respect to the WMO standard (Hearn et al.,
1961). This fulfills the GAW requirement.
The ozone data for all sites are stored at half-hour intervals. The times are specified for the end of the averaging
interval in Central European Time (CET, = UTC + 1 h). 1-h averages for the Zugspitze stations were made
available to the World Data Center and the TOAR data base (Schultz et al., 2017). In the present study we use
data at half-hour time resolution.
**2.6 LAGRANTO Trajectories**
Fifteen-day backward trajectories were calculated with the Lagrangian Analysis tool (LAGRANTO; Sprenger
and Wernli, 2015; Wernli and Davies 1997). The driving wind fields are obtained from the ERA5 reanalysis
dataset (Hersbach et al., 2020), which we interpolated to a 0.5° latitude/longitude grid, and on 137 vertical hybrid
levels. The input ERA5 data are available at a one-hour temporal resolution; the output positions of the
trajectories are written at 15-min time interval to allow for a more refined analysis. The starting coordinates of
the backward trajectories are 11.064 E, 47.477 N, and the starting altitudes match the altitudes of interest in the
soundings (see Sect. 4). The start times of the trajectories correspond to the sounding times within five minutes.
Finally, the start times are also shifted by several hours relative to the sounding time to assess the sensitivity of
the trajectory calculation on time.
**3. Results**
The main problem in comparing vertical sounding is illustrated in Fig. 1 which shows several ozone measure-
ments at Garmisch-Partenkirchen and Hohenpeißenberg in the morning of 2 October 2017. The vertical
distributions during that period are characterized by a descending stratospheric intrusion layer (see low relative
humidity) of rapidly diminishing width and significant changes at all altitudes on a short time scale. This reveals
the spatially highly inhomogeneous air mass. The approximate agreement of lidar and Hohenpeißenberg ozone
sonde before 6:00 CET is, thus, fortuitous. At different altitudes different air components must be assumed as
indicated by matching of the sonde ozone with lidar measurements at different times.
Until 2010 the lidar results were routinely compared with the long-term measurements at Wank and Zugspitze.
Apart from occasional orographically induced deviations an agreement to within ±2 ppb was found. After these
*in-situ* measurements were terminated we have routinely compared the lidar measurements with the ozone
measurements at UFS. Mostly a similar agreement is found.
**3.1 Comparisons of the IFU ozone lidar and the Jülich ECC sonde**
An optimum lidar validation became possible in early 2019. On 5 and 6 February 2019 a side-by-side instrument
comparison took place at Garmisch-Partenkirchen as a contribution to the FIRMOS (Far-Infrared Radiation
Mobile Observation System) validation project of the European Space Agency (Palchetti et al., 2021; Di Natale
et al., 2021). Two of the three balloons launched on 5 February were equipped with ozone sondes, while both
ballons on 6 February carried an ozone sonde. The ascents took place during night-time because of comparisons
of the CFH sondes with the water-vapour channel of the UFS Raman lidar that provides humidity profiles up to
at least 20 km (Klanner et al. 2021).



The first night of the campaign was clearer. The conditions for the comparison were excellent: the sondes rose
almost vertically up to 8.5 km and then slowly drifted to the south-east (Innsbruck), ideal for the comparison.
The balloons stayed within 20 km distance from IMK-IFU up to the tropopause (12.8 km a.s.l.) and remained
within 30 km up to 20 km a.s.l.
The launch times of the balloons were 18:03 CET (ascent to 16.147 km), 19:03 CET (29.475 km), and 23:00
CET (29.469 km).
In Fig. 2 we present the results of the four comparisons made. The measurements of lidar, ECC sonde and *in-situ*
sensor on 5 February are in outstanding agreement provided that a small constant offset is applied to the sonde
ozone between −0.53 and −3.4 ppb for the first three comparisons. The DIAL measurements are smoothed with a
numerical filter with an interval width growing with altitude (Trickl et al., 2020a). Nevertheless, the agreement
towards the tropopause is exceptional considering the low differential absorption for the wavelength pair 292
nm – 313 nm typically used above 6 km.
In addition, we show in Fig. 2 the results of three humidity measurements with the UFS Raman lidar (Klanner et
al., 2021) at a distance of 9 km from IMK-IFU. The water-vapour mixing ratios (MR) indicate a high variability
of the air composition between 5 and 8 km on both days, with a series of rapidly changing dry layers. In this
altitude range the MR do not agree quantitatively with those obtained with the CFH sondes. The MR of the lidar
is much less modulated because of the 1-h data-acquisition time necessary for the stratosphere. Because of this
variability the excellent air-mass matching by the side-by-side ozone soundings at IMK-IFU is crucial for the
results.
On 6 February the quality of the lidar retrievals was deteriorated by a layer of cirrus clouds above 9 km, which
required an aerosol correction. An increased level of ozone in this layer is remarkable, but is verified by the
sonde. By contrast, Reichardt et al. (1996) reported full ozone depletion in a cirrus layer that we traced back to
the surface of the Pacific Ocean where ozone destruction can be assumed to prevail (Kley et al., 1996). The
fourth comparison shows less perfect agreement because the lidar measurements ended at 19:00 CET, hours
before the last sonde ascent. This was the final measurement of the DIAL before its operation was terminated
after almost three decades.
Ozone profiles are also available for the descent of the balloons. The descents took place over Northern Italy and
intersected different air masses. As a consequence, strong discrepancies are seen, and we do not include these
data.
The first three comparisons yield average deviations of the sonde from the lidar. The average sonde offsets,
determined up to 5.8 km, were first subtracted from the sonde ozone values. Then, the differences between the
corrected sonde and the lidar data were formed at intervals of 52.5 m, for the first comparison on 6 February just
up to 8.7 km. Finally, these differences were averaged (Fig. 3). The agreement between the two systems without
the single-measurement offsets up to 9.2 km is ±2.5 ppb (about ±5 %). This is also the agreement we have found
between lidar and the mountain stations over the years and, thus, characterizes the winter-time performance of
the lidar after 2011.
The quality of the comparison shown in this section benefits from low to moderate ozone densities during the
cold season, which ensures limited absorption of the laser radiation within the troposphere. In Sect. 3.2 we assess
the performance for all seasons.



### 3.2 Comparison of MOHp ozone soundings with IFU lidar and *in-situ* measurements for 2018

The routine measurements with the IFU ozone DIAL exhibit rather different annual coverages, with gaps due to system damage or upgrading periods. Starting in late 2012 the final technical performance was reached. Retrieval strategies have been further improved. The best coverage of a single year was achieved in 2018 with a total of 587 measurements and 16 (March) to 79 (September) measurements per month. Therefore, we use this year for a thorough comparison with the MOHp ozone sonde.

The sonde ascents at MOHp usually take place around 6:00 CET on Monday, Wednesday and Friday, in summer just on Monday and Wednesday. We found a total of 46 of these days on which early-morning lidar measurements exist, not later than around 10:00 CET. On 36 of these days MOHp soundings are available. Thirteen of the days provided particularly good conditions with favourable temporal proximity. In the figures shown in this paper we eliminate ozone profiles for times later than 10:00 CET during a given day.

*Winter*

During the cold parts of the year the comparisons usually exhibit better quality. This is explained by less structured ozone vertical distributions and a wider operating range of the lidar due to the low ozone level allowing for a higher, less noisy far-field signal. This was already demonstrated in the previous section. For the 2018 comparison we give one example in Fig. 4. The lidar mixing ratio is of the order of 45 ppb, verified by the measurements at UFS (2660 m a.s.l.). The sonde results match the lidar values well if one adds 5.8 ppb. Just below the tropopause there is a minor discrepancy that could be either due to the higher uncertainty of the lidar measurement at these altitudes or air-mass differences.

*Summer*

During the warm season the ozone distribution in the middle and upper troposphere shows structured maxima caused by long-range transport, in particular STT (stratosphere-to-tropopause transport) layers (Trickl et al., 2020b). In this altitude range a summer maximum of STT exists. Usually, these structures do not perfectly match for both sites. An example for 9 July 2018 is shown in Fig. 5.

Figure 5 shows good agreement in structure between the soundings at both sites up to 9 km, in the presence of northerly advection. Again, the agreement was improved on the absolute scale by adding a correction to the sonde values (6 ppb). The elevated ozone between 3.3 km and 4.7 km can be explained by a stratospheric air intrusion, as is verified by the low RH. In the upper troposphere the agreement deteriorates, but at least the increase of ozone with altitude is seen in all profiles up to about 12 km. The ozone minimum around 13 km is just seen in the lidar data, with just a small ozone dip in the sonde profile. It is unreasonable to ascribe this considerable discrepancy to a temporary technical problem in such a limited altitude range. This example documents the difficulty of quantitative comparisons of tropospheric ozone even on a horizontal scale of just 38 km.

In order to clarify the origin of the difference of the ozone mixing ratio in the upper troposphere we calculated backward trajectories with the HYSPLIT model (http://ready.arl.noaa.gov/HYSPLIT.php; Draxler and Hess, 1998; Stein et al., 2015). These trajectories did not fully explain the observations due to the limited maximum backward time span of 315 h. This includes "ensemble" trajectory bundles that visualize a wider range of source regions.

Therefore, the trajectory calculations were extended to 350 h by using the LAGRANTO model for full-hour start times between 3:00 CET and 8:00 CET. Results for start times of 7:00 CET and 8:00 CET are shown in Figs. 6



and 7. Up to a start time of 4:00 CET the trajectories stayed almost completely at high altitudes. At 5:00 CET
three of the trajectories ended in the lower troposphere above the subtropical Pacific near a longitude of 180°,
first sign of an air-mass change. Later (Figs. 6 and 7) we see a clear influence of a Pacific source.
The low ozone level in the boundary layer above (sub)tropical oceans is well known (Eisele et al., 1999; Grant et
al., 2000; Trickl et al., 2003), in particular over the Pacific (Kley et al., 1996; Davies et al., 1998). In this way,
the lidar observations on 9 July 2018 can be understood. The launch time of the MOHp ozone sonde, 5:42 CET,
is between the two lidar measurements. However, a delay is caused by the northerly advection.
The moderate sonde RH indicates a potential admixture of aged stratospheric air above MOHp which would
explain the high ozone mixing ratios of more than 120 ppb.
Figures 5 and 8 show a rather constant negative ozone offset of the sonde profiles. The ozone profiles can be
brought into much better agreement by upward shifts by 6 ppb and 10 ppb, respectively. In Fig. 9 one sees one of
the very rare cases of a clear ozone mismatch between sonde and lidar up to elevations clearly above the
mountain sites (1 km above the Zugspitze summit). We conclude that differences between the Zugspitze sites
and the MOHp sonde are mostly not related to differences in air-composition in contrast to what was suspected
earlier.
The offsets of the MOHp data were evaluated for all 36 comparison days. The result is displayed in Fig. 10
where also the differences between the lidar results for 2671 m a.s.l. and the GAW measurements at UFS are
shown.
As found for the lidar measurements over many years (examples: Trickl et al., 2014, 2015, 2016, 2020b) the
lidar ozone agrees with that at UFS to within 2 to 3 ppb. The agreement would be better if orographic vertical
displacements and air flows on the ozone profiles would be considered (Carnuth et al., 2000; 2002; Yuan et al.,
2019). Orographic effects matter particularly in summer. Under warm conditions the lidar ozone seems to be
slightly higher on average with respect to UFS. A strong negative shift of −7 ppb can be seen in Fig. 5 where
UFS is located in the falling edge of a high-ozone range. This case was discarded from the statistical assessment.
The average difference between lidar and UFS for 2018 is 0.736 ppb ± 1.46 ppb (standard deviation). A positive
offset had also been found for an earlier four-day comparison with the Zugspitze summit, but with even higher
uncertainty (Trickl et al., 2020a). A positive offset of this size could be expected from the highly accurate cross-
section measurements of Viallon et al. (2015), who determined a negative bias of 1.8 % of the *in-situ* data
calibrated with the WMO standard. This relative difference becomes more important on the absolute scale in
summer than in winter because of the higher ozone values. It might explain the slight seasonal cycle of the
difference visible in Fig. 10. However, given the complex orography at the site we think that the uncertainties are
too high for such a conclusion.
The offsets between the MOHp sonde and the lidar are substantially higher. We exclude the lowest altitudes
from the comparison where obvious differences in ozone exist, e.g., due to local night-time ozone depletion
effects. The comparisons with the Zugspitze summit are mostly reasonable: Just in seven cases of the 36
comparisons for 2018 lower ozone in the sonde profiles reached up to more than 2.67 km (UFS), in three cases
to more than 3 km (Zugspitze summit). The offsets of the ozone sondes range from −12 ppb to + 4 ppb, with an
average of −3.77 ppb and a standard deviation of 4.22 ppb.
***Differences***
In Figs. 11 to 13 we show average differences between lidar and offset-corrected MOHp sonde data as a function
of altitude and for three different ozone conditions, roughly below 50 ppb (low ozone), between 50 to 70 ppb



(moderate ozone) and more than 70 ppb (high ozone). The averaging was carried out just for measurement days
with lidar measurements in temporal proximity to the launch time of the ozone sonde. We also give the
percentages of the averages with respect to the offset-corrected sonde ozone. At high altitudes the sonde ozone is
a more useful reference than the lidar in the case of high ozone because of the considerable absolute uncertainty
caused by the loss of laser radiation.
For winter-type conditions (mixing ratio mostly less than 50 ppb) the six examples averaged exhibit low vertical
ozone structure which made the analysis straight forward and yields astonishingly small average differences
between 1 and 3 ppb. For moderate ozone and high ozone, mostly during the warm season, the vertical
distributions are more complex with changes on a time scale of even less than one hour. Here, we eliminated
several obvious ozone peaks and dips that differed at both stations. The six high-ozone cases were restricted to
July and August.
The averaged distributions of the differences exhibit oscillations. These oscillations were analysed for coherency
(not shown), but no systematic behaviour was identified. Thus, we ascribe the structure to noise. The noise
contains both an atmospheric and an instrumental component.
The noise amplitude shrinks above 6 km because of the change from 277 nm as the "on" wavelength to 292 nm.
This step is not clearly seen in Fig. 13 due to the higher 292-nm noise level. In July and August there are cases
with 100 to 150 ppb in the middle and upper troposphere. This can lead to lidar uncertainties even up to more
than 20 ppb during day-time because of the additional solar background noise. This is larger than the excursions
in the average in Fig. 13.
The analyses for 2018 do not reveal a significant bias between the lidar values and the offset-corrected sonde
data (based on the numbers underlying Fig. 10). The maximum noise excursions can be interpreted as maximum
combined uncertainties of lidar and sonde in a given altitude range. The results of this analysis confirm the
estimates in Table 4 of Trickl et al. (2020a).
**3.3 Comparisons of MOHp sonde, IFU lidar and *in-situ* measurements summits in 2009**
The results in Sect. 3.2 suggested to look also at a few earlier years. We select 2009 from the period of routine
measurements as another year of comparison. The lidar raw data were noisier than for the period after 2012 and
a tiny electronic ringing effect had to be removed mathematically. Thus, the uncertainties of the ozone profiles
above 6 km are higher than after the final system upgrading in 2012, particularly in summer. As a consequence,
a lidar validation is desirable at least for the upper troposphere. More importantly, in 2009 high-quality ozone
data still exist for the summit stations Wank (1780 m a.s.l.) and Zugspitze (2962 m a.s.l.). These stations benefit
from more frequent direct advection compared with UFS.
In 2009 the lidar was operated just until October which, nevertheless, allows us to make a reasonable number of
comparisons with MOHp. The operation was stopped afterwards since there were more and more cases of single-
bit errors in channel 5 of the transient digitizer system which had to be sent for repair. These errors induced
unrealistic data in the upper troposphere.
We identified a total of 23 days suitable for comparisons. On just eight of these days lidar measurements were
made in optimum temporal proximity. We find more deviations in the profiles than for 2018. In part, this can be
explained by atmospheric variability and insufficient air-mass matching. In addition, as mentioned, the raw data
of the lidar are noisier and some weak ringing had to be removed. This caused elevated uncertainties above 6
km. Nevertheless, the data allowed us to determine offsets for the MOHp ozone profiles, after verifying the data
quality of the lidar with the Zugspitze and Wank *in-situ* ozone.





In Fig. 14 we show the results of the analysis for 2009. The difference between IFU DIAL and Zugspitze is
−0.165 ppb ± 1.36 ppb (standard deviation), between DIAL and Wank +0.714 ppb ± 1.20 ppb. The DIAL ozone
below the Wank altitude is increasingly uncertain because of alignment issues of the near-field telescope. In an
earlier comparison for May 1999 (Trickl et al., 2020a) we selected a lower altitude in the DIAL data (2786 m)
and found better agreement, but, still, a slight positive offset with respect to the station. This is not attempted
here, although we can see the effect of orographic lifting in some examples.
For 2009 the offsets between DIAL and MOHp sondes were determined primarily by between 2 and 5 km. The
sonde offset obtained in this way is, again, negative on average (−1.500 ppb), with a standard deviation of 2.67
ppb, both being are less pronounced than in 2018.
Figure 15 shows a comparison on 12 January 2009, demonstrating excellent agreement between both systems,
except for the upper troposphere and lower stratosphere. In this case, the first lidar measurement took place at
9:20 CET, i.e., substantially later than the sonde ascent. Thus, the comparison has its limits. In the morning of 12
January westerly advection was revealed by HYSPLIT backward trajectories above at 7 km a.s.l.. This air mass
originated below 2 km over the subtropical. This could explain the slightly lower ozone level around this altitude
in the lidar results.
Another interesting example is August 17 (Fig. 16). The agreement between lidar and ozone sonde is highly
satisfactory up to 5.4 km and quite reasonable up to 10 km. However, between 10 km and 14.5 km the lidar
ozone is extremely low, in contrast to the sonde data. The pronounced ozone increase in the sonde data above 10
km is difficult to explain since the elevated RH values suggest neither a low tropopause nor the presence of a
stratospheric intrusion. On the other hand, the ozone peak above IMK-IFU descending roughly from 10 to 8 km
is attributed by HYSPLIT calculations to subsiding air, indicating the presence of an intrusion layer. It is
interesting that the rather short delay of the lidar measurements (7:00 CET to 9:15 CET) with respect to the
sonde ascent (launch time 5:57 CET) can result in such a considerable difference.
Again, 350-h LAGRANTO trajectories were calculated for start times above IMK-IFU between 3:00 CET and
8:00 CET (interval: 1 h) and start altitudes within the low-ozone layer. Until 6:00 CET the influence of marine
boundary layers is almost absent. Afterwards, the trajectories reveal a growing import from the first 600 m above
the subtropical Atlantic Ocean. In Fig. 17 the LAGRANTO results for 8:00 CET are shown.
In many cases the lidar seems to exhibit a negative bias with respect to the sondes in the upper troposphere. It is
advisable to re-examine a major part of the data between 2007 and 2011, also including strategies developed
later. For example, an exponential decay of the analogue signal was identified with the much lower noise of the
final setup (Trickl et al., 2020a) which must be addressed.

**3.4 Comparisons of MOHp sonde, IFU lidar and *in-situ* measurements summits in 2000 and 2001**

The period September 2000 to August 2001 is suitable for another comparison when a large number of STT-
related measurement series were made as a contribution to the STACCATO project (Stohl et al., 2003;
examples: Trickl et al., 2003; 2010; 2011; Zanis et al., 2003). These measurements were made with the detection
electronics of Eisele et al. (1999), but had the advantage that single-photon counting was used for the "solar
blind" "on" detection channels which added linearity above 5 km (starting in spring 1997). The counting system
was abandoned after 2003. A new one was installed after highly positive results in other IFU lidar systems in
2018 (Klanner et al., 2021), too late for routine operation.
The focus on STT during the STACCATO period made the comparisons a challenge because of the pronounced
layering. However, on 11 of the useful 20 days of comparison there was reasonable temporal proximity, due to
running long time series. The agreement between the lidar and the MOHp sonde was much better than expected
in the entire free troposphere. The agreement (after offset-correcting the MOHp profiles) is almost perfect during
the cold season. But also under high-ozone conditions the comparisons do not reveal systematic differences
beyond the sonde offsets.
Two examples for elevated ozone are shown in Figs. 18 and 19. The good comparisons support our earlier work
(Trickl et al., 2003, and 2010, respectively), and we tend to ascribe this to the good performance of the single-
photon counting system.
For several weeks a strange ozone rise towards the ground was observed in the lidar data below 1.5 km. This
effect disappeared after realigning the near-field telescope and the normal early-morning ozone drop returned.
However, the offsets of the MOHp mixing ratios necessary to achieve good agreement are, again, quite
substantial (Fig. 20). Also the differences between lidar and the stations are higher than those in the preceding
sections, and comparable with those of the mentioned four-day comparison for May 1999 (Trickl et al., 2020a).
The statistical analysis yields the following average differences and standard deviations:
IFU DIAL – Zugspitze:     1.22 ppb ± 1.81 ppb
IFU DIAL – Wank      −0.15 ppb ± 2.26 ppb
MOHp – IFU DIAL     −5.88 ppb ± 3.35 ppb
**4. Discussion and Conclusions**
For some time tropospheric differential-absorption ozone lidar systems had a bad reputation: The method is
highly sensitive to imperfections in the signal acquisition since the ozone number density is obtained by
derivative formation. In addition, a lidar covering the entire troposphere and the lowermost stratosphere features
a dynamic range of the backscatter signal of about eight decades, which means an extreme challenge for the
detection electronics.
Based on continual improvements, starting with the 1994 system upgrading, the IFU ozone DIAL gradually
approached a high performance until 2012, but there is, still, minor potential for improvements. Comparison
with the nearby mountain stations quite early demonstrated an uncertainty level of ±3 ppb in the lower
troposphere. Occasional comparisons with ozone sondes launched at the Hohenpeißenberg (1996 to 2001,
distance 38 km) were rather satisfactory up to the tropopause region.
Here, we more comprehensively analyse the lidar performance during three periods in its technical development.
The best agreement was found for the side-by-side comparison with balloon ascents of ozone sondes, performed
by the FZJ team at IMK-IFU in February 2019. Just a small, constant offset had to be subtracted from the sonde
data to achieve perfect agreement. For the more distant MOHp sonde the comparisons are more demanding
because of the high atmospheric variability (Vogelmann et al., 2011; 2015). This variability is particularly severe
in summer when the atmospheric layering is more pronounced. Nevertheless, there was enough agreement in
certain altitude ranges for examining the reliability of the ozone profiles obtained from the DIAL, also before the
final modifications in 2012. It turned out that just between 2007 and 2011 we can suspect a slight negative
summertime bias of the lidar of the order of 5 ppb above 6 km. This could be due to interfering structures on the
analogue signal (requiring mathematical correction) that could not be compensated by photon counting
(available just until 2003). In principle, this calls for a re-evaluation of the ozone profiles over the period 2007 to
2011, based on more recent experience in the signal inversion and the performance of the electronic equipment.
Vice versa, the lidar measurements helped us to identify the quality of the sonde measurements. Quite good
agreement can be achieved by applying an altitude-independent offset correction to the ozone values that





strongly varies from sonde to sonde. In all but a few cases the offset can be determined to within ±2 ppb by a
comparison with the Zugspitze or the UFS station data. Most of the time differences in early-morning air-mass
composition between the two sites are limited to altitudes below 2 km. Thus, the differences reported earlier by
Scheel for 3 km (see introduction) are not caused by differences in air composition at both sites. A comparison
with the *in-situ* data is advisable despite the considerable distance between the sites.
The comparisons for the three years 2000-2001, 2009 and 2018 reveal just minor performance change of the
MOHp sonde over the years, with a variation of the annual average offset by about ±2 ppb. We found a negative
average offset of −3.64 ppb ± 3.72 ppb (standard deviation) with respect to the IFU ozone DIAL over all three
years. It is reasonable to assume that this offset is applicable to the entire tropospheric time series of the MOHp
sondes.
Remaining tasks for the lidar are a substantial reduction of the solar background at 313.2 nm in summer and to
enhance the moderate 291.8-nm backscatter signal in the upper troposphere. Further reduction of the residual
solar background is difficult since the spectral filtering is already quite narrow. However, replacement of the
rather aged (and partly contaminated) primary mirror of the far-field receiver could help by reducing the
background radiation reflected into the detection system. As mentioned longer averaging is advisable. By longer
averaging, the performance under low-aerosol conditions could almost reach that of *in-situ* measurements in a
major part of the troposphere. Single-photon counting can also be helpful for longer averaging times, as
demonstrated for our Raman lidar (Klanner et al., 2021). The noise level for counting is still lower than that of
the meanwhile outstanding transient digitizers (Trickl et al., 2020a).
**5 Data availability**
Lidar data and information on the lidar systems can be obtained on request from the IMK-IFU authors of this
paper (thomas@trickl.de, hannes.vogelmann@kit.edu). The 313-nm aerosol backscatter coefficients are archived
in the EARLINET data base, accessible through the ACTRIS data portal http://actris.nilu.no/. The
Hohenpeißenberg ozone and humidity data are stored in the NDACC data archive (https://www-
air.larc.nasa.gov/missions/ndacc/data.html#). The data of the FIRMOS campaign is available via the ESA
campaign dataset website https://earth.esa.int/eogateway/campaigns/firmos.
**6 Author statement**
TT carried out most lidar measurements after spring 1997, following U. Kempfer and H. Eisele. He led the
technical development of two ozone DIAL systems since 1990. HV was involved in the system upgrading since
2007 and was responsible for the lidar operation during FIRMOS. DC and CW launched several ECC sondes at
IMK-IFU in February 2019. MA and WS carried out the MOHp ozone sonde measurements. LR performed
Ozone *in-situ* measurements at the site Schneefernerhaus. MS provided LAGRANTO backward trajectories.
**7 Competing interests**
The authors declare that they have no conflict of interest.
**Acknowledgements**
The authors thank Wolfgang Seiler and Hans Peter Schmid for their support over so many years. The late Hans-
Eckhart Scheel provided reference ozone data for the Wank, Zugspitze and Schneefernerhaus mountain stations
in the vicinity of IMK-IFU The different steps of lidar development have been funded by the German Ministry

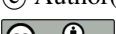



of Research and Technology (BMFT), the German Foundation for the Environment (DBU, two projects), and the
Bavarian Ministry of Economics. Since 2007 the 313-nm aerosol results have contributed to EARLINET
(European Aerosol Research Lidar Network) that is currently a part of the European Research Infrastructure
ACTRIS. The lidar measurements were also funded by the European Union within Vertical Ozone Transport 1
and 2 (e.g., Wotava, G., and Kromp-Kolb, 2000; VOTALP 2, 2000) and STACCATO (Stohl et al., 2003) and by
the German Ministry for Research and Education (BMBF) within ATMOFAST (2005).
KIT acknowledges support of lidar measurements by the European Space Agency (ESA) under Contract
4000123691/18/NL/NF (FIRMOS validation campaign). Balloon profiles utilized in this paper have been
provided within the same ESA project by the Forschungszentrum Jülich via subcontract with KIT. The balloon
activities were also partly supported by the Helmholtz Association in the framework of MOSES (Modular
Observation Solutions for Earth Systems).
The service charges for this open access publication have been covered by a Research Centre of the Helmholtz
Association.

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

Methods to homogenize electrochemical concentration cell (ECC) ozonesonde measurements across changes in
sensing solution concentration or ozonesonde manufacturer, Atmos. Meas. Tech., 10, 2021–2043, 2017.
Daumont, D., Brion, J., Charbonnier, J., and Malicet, J.: Ozone UV Spectroscopy I: Absorption Cross-Sections
at Room Temperature, J. Atmos. Chem., 15, 145-155, 1992.
Davies, W. E., Vaughan, G., and O´Connor, F. M.: Observation of near-zero ozone concentrations in the upper
troposphere at mid-latitudes, Geophys. Res. Lett., 25, 1173-1176, 1998.
De Backer, H., De Muer, D., and De Sadelaer G.: Comparison of ozone profiles obtained with Brewer-Mast and
Z-ECC sensors during simultaneous ascents, J. Geophys. Res., 103, 19,641–19,648,
https://doi.org/10.1029/98JD01711 , 1998.
De Muer, D., and Malcorps, H.: The frequency response of an electrochemical ozone sonde and its application to
the deconvolution of ozone profiles, J. Geophys. Res., 89, 1361–1372, 1984.
Di Natale, G., Barucci, M., Belotti, C., Bianchini, G., D'Amato, F., Del Bianco, S. Gai, M., Montori, A.,
Sussmann, R., Viciani, S., Vogelmann, H., and Palchetti, L.: Comparison of mid-latitude single- and mixed-
phase cloud optical depth from co-located infrared spectrometer and backscatter lidar measurements, Atmos.
Meas. Tech., 14, 6749–6758, 2021.
Dirksen, R. J., Sommer, M., Immler, F. J., Hurst, D. F., Kivi, R., and Vömel, H.: Reference quality upper-air
measurements: GRUAN data processing for the Vaisala RS92 radiosonde, Atmos. Meas. Tech., 7, 4463-4490,
doi.org/10.5194/amt-7-4463-2014, 2014.
Draxler, R., and Hess, G.: An overview of the HYSPLIT_4 modelling system for trajectories, dispersion, and
deposition, Aust. Meteorol. Mag., 47, pp. 295-308, 1998.
Eisele, H., Trickl, T., and Claude, H.: Lidar als wichtige Ergänzung zur Messung troposphärischen Ozons,
Ozonbulletin des Deutsches Wetterdiensts, 44, 2 pp., 1997 (in German).
Eisele, H., Scheel, H. E., Sladkovic, R., and Trickl, T.: High-Resolution Lidar Measurements of Stratosphere-
Troposphere Exchange, J. Atmos. Sci., 56, 319-330, 1999.
Eisele, H., and Trickl, T.: Improvements of the aerosol algorithm in ozone-lidar data processing by use of
evolutionary strategies, Appl. Opt., 44, 2638-2651, 2005.
EUROTRAC: Transport and Chemical Transformation of Pollutants in the Troposphere, Vol. 1, An Overview of
the Work of EUROTRAC, P. Borrell and P. M. Borrell, Eds., Springer (Berlin, Heidelberg, New York), ISBN 3-
540-66775-X, 474 pp., 1997.
Gaudel, A., Ancellet G., and Godin-Beekmann S.: Analysis of 20 years of tropospheric ozone vertical profiles by
lidar and ECC at Observatoire de Haute Provence (OHP) at 44° N, 6.7° E, Atmos. Environ., 113, 78-89, 2015.


Gaudel, A., Cooper, O. R., Ancellet, G., Barret, B., Boynard, A., Burrows, J. P., Clerbaux, J. P., Coheur, P.-F., Cuesta, J., Cuevas, E., Doniki, S., Dufour, G., Ebojie, F., Foret, G., Garcia, O., Granados-Muñoz, M. J., Hannigan, J., Hase, F., Hassler, B., Huang, G., Hurtmans, D., Jaffe, D., Jones, N., Kalabokas, P., Kerridge, B., Kulawik, S., Latter, B., Leblanc, T., Le Flochmoën, E., Lin, W., Liu, J., Liu, X., Mahieu, E., McClure-Begley, A., Neu, J., Osman, M., Palm, M., Petetin, H., Petropavlovskikh, I., Querel, R., Rahpoe, N., Rozanov, A., Schultz, M. G., Schwab, J., Siddans, R., Smale, D., Steinbacher, M., Tanimoto, H., Tarasick, D., Thouret, V., Thompson, A., M., Trickl, T., Weatherhead, E., Wespes, C., Worden, H., Vigouroux, C., Xu, X., Zeng, G., Ziemke, J.: Tropospheric Ozone Assessment Report: Present-day distribution and trends of tropospheric ozone relevant to climate and global atmospheric chemistry model evaluation, Elem. Sci. Anth., 6, 39, DOI: https://doi.org/ 10.1525/elementa.291, 58 pp., 2018.

Grant, W. B., Browell, E. V., Butler, C. F., Fenn, M. A., Clayton, M. B., Hannan, J. R., Fuelberg, H. E., Blake, D. R., Blake, N. J., Gregory, G. L., Heikes, B. G., Sachse, G. W., Singh, H. B., Snow, J., and Talbot, R. W.: A case study of transport of tropical marine boundary layer and lower tropospheric air masses to the northern midlatitude upper troposphere, J. Geophys. Res., 105, 3757-3769, 2000.

Hearn, A. G.: The Absorption of Ozone in the Ultra-violet and Visible Regions of the Spectrum, Proc. Phys. Soc., 78, 932-940, 1961.

Hersbach, H, Bell, B, Berrisford, P, Simmons, A., Berrisford, P., Dahlgren, P., Horanyi, A., Munoz-Sabater, J., Nicolas, J., Radu, R., Schepers, D., Soci, C., Villaume, S., Bidlot, J. R., Haimberger, L.; Woollen, J., Buontempo, C., and Thepaut, J. N.: The ERA5 global reanalysis. Q. J. R. Meteorol Soc., 146, 1999– 2049, https://doi.org/10.1002/qj.3803, 2020.

Johnson, B. J., Oltmans, S. J., Vömel, H., Smit, H. G. J., Deshler, T., and Kröger, C.: Electrochemical concentration cell (ECC) ozonesonde pump efficiency measurements and tests on the sensitivity to ozone of buffered and unbuffered ECC sensor cathode solutions, J. Geophys. Res.-Atmos., 107, ACH 8-1–ACH 8-18, https://doi.org/10.1029/2001JD000557, 2002.

Jeannet, P., Stübi, R., Levrat, G., Viatte, P., and J. Staehelin, J.: Ozone balloon soundings at Payerne (Switzerland): Reevaluation of the time series 1967–2002 and trend analysis, J. Geophys. Res., 112, D11302, doi:10.1029/2005JD006862, 15 pp., 2007.

Kempfer, U., Carnuth, W., Lotz, R., and Trickl, T.: A wide range ultraviolet lidar system for tropospheric ozone measurements: development and application, Rev. Sci. Instrum., 65, 3145-3164, 1994.

Kerr, J. B., Fast, H., McElroy, C.T., Oltmans, S.J., Lathrop, J.A., Kyro, E., Paukkunen, A., Claude, H., Köhler, U., Sreedharan, C.R., akao T., and Tsukagoshi, Y.: The 1991 WMO International Ozonesonde Intercomparison at Vanskoy, Canada. Atmos.–Ocean, 32, 685–716, https://doi.org/10.1080/07055900.1994.9649518, 1994.

Klanner, L., Höveler, K., Khordakova, D., Perfahl, M., Rolf, C., Trickl, T., and Vogelmann, H.: A powerful lidar system capable of 1 h measurements of water vapour in the troposphere and the lower stratosphere as well as the temperature in the upper stratosphere and mesosphere, Atmos. Meas. Tech., 14, 531-555, 2021.

Klausen, J., Zellweger, C., Buchmann, B., and Hofer, P.: Uncertainty and bias of surface ozone measurements at selected Global Atmospheric Watch sites, J. Geophys. Res., 108, 4622, doi: 10.1029/2003JD003710, 17 pp., 2003.



Kley, D., Beck, J., Grennfelt, P. I., Hov, O., and Penkett, S. A.: Tropospheric Ozone Research (TOR) A Sub-
Project of EUROTRAC, J. Atmos. Chem., 28, 1–9, 1997.
Kley, D., Crutzen, P. J., Smit, H. G. J., Vömel, H., Oltmans, S., Grassl, H., and Ramanathan, V.: Observations of
Near-Zero Ozone Concentrations Over the Convective Pacific: Effects on Air Chemistry, Science, 274, 230-233,
647    1996.

Komhyr, W.D.: Electrochemical concentration cells for gas analysis, Ann. Geoph., 25, 203–210, 1969.
Komhyr, W. D., Barnes, R. A., Brothers, G. B., Lathrop, J. A., and Opperman, D. P.: Electrochemical
concentration cell ozonesonde performance evaluation during STOIC 1989, J. Geophys. Res., 100, 9231–9244,
https://doi.org/10.1029/94JD02175, 1995.
Langford, A. O., Masters, C. D., Proffitt, M. H., Hsie, E.-Y., and Tuck, A. F.: Ozone measurements in a
tropopause fold associated with a cut-off low system, Geophys. Res. Lett., 23, 2501–2504, 1996.
Logan, J.A., Staehelin, J., Megretskaia, I.A., Cammas, J.-P., Thouret, V., Claude, H., De Backer, H.,
Steinbacher, M., Scheel, H.-E., Stübi, R., Fröhlich, M., and Derwent, R. (2012), Changes in ozone over Europe:
Analysis of ozone measurements from sondes, regular aircraft (MOZAIC) and alpine surface sites, J. Geophys.
Res., 117, D09301, doi:10.1029/2011JD016952.
Malicet, J., Daumont, D., Charbonnier, J., Parisse, C., Chakir, A., and Brion, J.: Ozone UV Spectroscopy I:
Absorption Cross-Sections and Temperature Dependence, J. Atmos. Chem., 21, 263-273, 1995.
Palchetti, L., Barucci, M., Belotti, C., Bianchini, G., Cluzet, B., D'Amato, F., Del Bianco, S., Di Natale, G., Gai,
M., Khordakova, D., Montori, A., Oetjen, H., Rettinger, M., Rolf, C., Schuettemeyer, D., Sussmann, R., Viciani,
S., Vogelmann, H., and Wienhold, F. G.: Observations of the downwelling far-infrared atmospheric emission at
the Zugspitze observatory, Earth Syst. Sci. Data, 13, 4303–4312, https://doi.org/10.5194/essd-13-4303-2021,
664    2021.

Schultz, M. G., Schröder, S., Lyapina, O., Cooper, O., Galbally, I., Petropavlovskikh, I., von Schneidemesser,
E., Tanimoto, H., Elshorbany, Y., Naja, M., Seguel, R. J., Dauert, U., Eckhardt, P., Feigenspan, S., Fiebig, M.,
Hjellbrekke, A.-G., Hong, Y.-D., Kjeld, P. C., Koide, H., Lear, G., Tarasick, D., Ueno, M., Wallasch, M.,
Baumgardner, D., Chuang, M.-T., Gillett, R., Lee, M., Molloy, S., Moolla, R., Wang, T., Sharps, K., Adame, J.
A., Ancellet, G., Apadula, F., Artaxo, P., Barlasina, M. E., Bogucka, M., Bonasoni, P., Chang, L., Colomb, A.,
Cuevas-Agulló, E., Cupeiro, M., Degorska, A., Ding, A., Fröhlich, M., Frolova, M., Gadhavi, H., Gheusi, F.,
Gilge, S., Gonzalez, M. Y., Gros, V., Hamad, S. H., Helmig, D., Henriques, D., Hermansen, O., Holla, R.,
Hueber, J., Im, U., Jaffe, D. A., Komala, N., Kubistin, D., Lam, K.-S., Laurila, T., Lee, H., Levy, I., Mazzoleni,
C., Mazzoleni, L. R., McClure-Begley, A., Mohamad, M., Murovec, M., Navarro-Comas, M., Nicodim, F.,
Parrish, D., Read, K. A., Reid, N., Ries, L., Saxena, P., Schwab, J. J., Scorgie, Y., Senik, I., Simmonds, P.,
Sinha, V., Skorokhod, A. I., Spain, G., Spangl, W., Spoor, R., Springston, S. R., Steer, K., Steinbacher, M.,
Suharguniyawan, E., Torre, P., Trickl, T., Weili, L., Weller, R., Xiaobin, X., Xue, L., and Zhiqiang, M.:
Tropospheric Ozone Assessment Report: Database and Metrics Data of Global Surface Ozone Observations,
Elem. Sci. Anth., 5, 58, DOI: https://doi.org/ 10.1525/elementa.244, 25 pp., 2017.
Reichardt, J., Ansmann, A., Serwazi, M., Weitkamp, C., and Michaelis, W.: Unexpectedly low ozone
concentration in midlatitude tropospheric ice clouds: A case study, Geophys. Res.Lett., 23, 1929-1932, 1996.



Smit, H. G. J, Straeter, W., Johnson, B. J., Oltmans, S. J., Davies, J., Tarasick, D. W., Hoegger, B., Stubi, R.,
Schmidlin, F. J., Northam, T., Thompson, A. M., Witte, J. C., Boyd, I., and Posny, F.: Assessment of the
Performance of ECC-ozonesondes under Quasi-flight Conditions in the Environmental Simulation Chamber:
Insights from the Jülich Ozone Sonde Intercomparison Experiment (JOSIE), J. Geophys. Res., 112, D19306,
doi:10.1029/2006JD007308, 18 pp., 2007.
Smit, H.G.J., and ASOPOS panel: Quality assurance and quality control for ozonesonde measurements in GAW,
World Meteorological Organization, GAW Report No. 201, Geneva (Switzerland). [Available online at
https://library.wmo.int/doc_num.php?explnum_id=7167], 100 pp., 2014.
Smit, H.G.J., and Thompson, A.M.: Ozonesonde Measurement Principles and Best Operational Practices:
ASOPOS 2.0 (Assessment of Standard Operating Procedures for Ozonesondes), World Meteorological
Organization, GAW Report No. 268, Geneva (Switzerland). [Available online at
https://library.wmo.int/doc_num.php?explnum_id=10884], 172 pp., 2021.
Stauffer, R.M., Thompson, A.M., Kollonige, D.E., Tarasick, D.W., Van Malderen, R., Smit, H.G.J., Vömel, H.,
Morris, G.A., Johnson, B.J., Cullis, P.D., Stübi, R., Davies, J., and Yan.: An Examination of the Recent Stability
of Ozonesonde Global Network Data, Earth and Space Science, 9 (10), e2022EA002459, [available online at
https://doi.org/10.1029/2022EA002459], 2022.
Stein, A. F., Draxler, R. R, Rolph, G. D., Stunder, B. J. B., Cohen, M. D., and Ngan, F.: NOAA's HYSPLIT
atmospheric transport and dispersion modeling system, Bull. Amer. Meteor. Soc., 96, 2059-2077, 2015.
Steinbrecht, W., Schwarz, R., and Claude, H.: New pump correction for the Brewer-Mast ozone sonde:
determination from experiment and instrument intercomparisons, J. Atmos. Ocean. Tech., 15, 144–156, 1998.
Stohl, A., and Trickl, T.: A textbook example of long-range transport: Simultaneous observation of ozone
maxima of stratospheric and North American origin in the free troposphere over Europe, J. Geophys. Res., 104,
703 30445-30462, 1999.

Stohl, A., Bonasoni, P., Cristofanelli, P., Collins, W., Feichter, J., Frank, A., Forster, C., Gerasopoulos, E.,
Gäggeler, H., James, P., Kentarchos, T., Kromp-Kolb, H., Krüger, B., Land, C., Meloen, J., Papayannis, A.,
Priller, A., Seibert, P., Sprenger, M., Roelofs, G. J., Scheel, H. E., Schnabel, C., Siegmund, P., Tobler, L., Trickl,
T., Wernli, H., Wirth, V., Zanis, P., and Zerefos, C.: Stratosphere-troposphere exchange - a review, and what we
have learned from STACCATO, J. Geophys. Res., 108, 8516, doi:10.1029/2002JD002490, STA 1, 15 pp., 2003.
Stübi, R., Levrat, G., Hoegger, B., Pierre Viatte, P., Staehelin, J., Schmidlin, F.J.: In-flight comparison of
Brewer-Mast and electrochemical concentration cell ozonesondes, J. Geophys. Res., 113, D13302,
https://doi.org/10.1029/2007JD009091 , 2008.
Tarasick, D. W., Davies, J., Anlauf, K., Watt, M., Steinbrecht, W., Claude H.-J.: Laboratory investigations of the
response of Brewer-Mast ozonesondes to tropospheric ozone, J. Geophys. Res., 107, ACH 14-1 – 14-10,
https://doi.org/10.1029/2001JD001167 , 2002.
Tarasick, D. W., Davies, J., Smit, H. G. J., and Oltmans, S. J.: A re-evaluated Canadian ozonesonde record:
measurements of the vertical distribution of ozone over Canada from 1966 to 2013, Atmos. Meas. Tech., 9, 195–
214, https://doi.org/10.5194/amt-9-195-2016, 2016.
Tarasick, D., Galbally, I. E., Cooper, O. R., Schultz, G M., Ancellet, G., Leblanc, T., Wallington, T. J., Ziemke,
J., Liu, X., Steinbacher, M., Staehelin, J., Vigouroux, C., Hannigan, J., García, O., Foret, G., Zanis, P.,





Weatherhead, E., Petropavlovskikh, I., Worden, H., Osman, M., Liu, J., Chang, K.-L., Gaudel, A., Lin, M.,
Granados-Muñoz, M., Thompson, A. M., Oltmans, S. J., Cuesta, J., Dufour, G., Thouret, V., Hassler, B., Trickl,
T., and Neu, J. L.: Tropospheric Ozone Assessment Report: Tropospheric ozone from 1877 to 2016, observed
levels, trends and uncertainties, Elem. Sci. Anth., **7**, Article 39, DOI: https://doi.org/10.1525/elementa.376, 72
pp. (plus 56 pp. of supplemental material), 2019.
Tarasick, D. W., Smit, H. G. J., Thompson, A. M., Morris, G. A., Witte, J. C., Davies, J., Davies, J., Nakano, T.,
Van Malderen, R., Stauffer, R. M., Johnson, B. J., Stubi1, R., Oltmans, S. J., and Vömel, H.: Improving ECC
ozonesonde data quality: Assessment of current methods and outstanding issues, Earth and Space Science, 8,
e2019EA000914. https://doi.org/10.1029/2019EA000914, 27 pp., 2021.
TESLAS: Tropospheric Environmental Studies by Laser Sounding (TESLAS), in: Transport and Chemical
Transformation of Pollutants in the Troposphere, Vol. 8, Instrument Development for Atmospheric Research and
Monitoring, J. Bösenberg, D. Brassington, and P. C. Simon, Eds., Springer (Berlin, Heidelberg, New York),
ISBN 3-540-62516-X, 1-203, 1997.
Thompson, A. M., Smit, H. G. J., Witte, J. C., Stauffer, R. M., Johnson, B. J., Morris, G., von der Gathen, P.,
Van Malderen, R., Davies, J., Piters, A., Allaart, M., Posny, F., Kivi, R., Cullis, P., Hoang Anh, N. T., Corrales,
E., Machinini, T., da Silva, F. R., Paiman, G., Thiong'o, K., Zainal, Z., Brothers, G. B., Wolff, K. R., Nakano,
T., Stübi, R., Romanens, G., Coetzee, G. J. R., Diaz, J. A., Mitro, S., Mohamad, M., and Ogino, S.: Ozonesonde
Quality Assurance: The JOSIE–SHADOZ (2017) Experience, Bulletin of the American Meteorological Society,
738 100, 155-171, 2019.

Trickl, T., Cooper, O. R., Eisele, H., James, P., Mücke, R., and Stohl, A.: Intercontinental transport and its
influence on the ozone concentrations over central Europe: Three case studies, J. Geophys. Res., 108, D12, 8530,
10.1029/2002JD002735, STA 15, 23 pp., 2003.
Trickl, T., Feldmann, H., Kanter, H.-J., Scheel, H. E., Sprenger, M., Stohl, A., and Wernli, H.: Deep
stratospheric intrusions over Central Europe: case studies and climatological aspects, Atmos. Chem. Phys., 10,
744 499-524, 2010.

Trickl, T., Eisele, H., Bärtsch-Ritter, N., Furger, M., Mücke, R., Sprenger, M., and Stohl, A.: High-ozone layers
in the middle and upper troposphere above Central Europe: potential import from the stratosphere along the
subtropical jet stream, Atmos. Chem. Phys., 11, 9343-9366; 5-p. Supplement, 2011.
Trickl, T., Vogelmann, H., Giehl, H., Scheel, H. E., Sprenger, M., and Stohl, A.: How stratospheric are deep
stratospheric intrusions? Atmos. Chem. Phys., 14, 9941-9961, 2014.
Trickl, T., Vogelmann, H., Flentje, H., and Ries, L.: Stratospheric ozone in boreal fire plumes – the 2013 smoke
season over Central Europe, Atmos. Chem. Phys., 15, 9631-9649, 2015.
Trickl, T., Vogelmann, H., Fix, A., Schäfler, A., Wirth, M., Calpini, B., Levrat, G., Romanens, G., Apituley, A.,
Wilson, K. M., Begbie, R., Reichardt, J., Vömel, H. and Sprenger, M.: How stratospheric are deep stratospheric
intrusions into the troposphere? LUAMI 2008, Atmos. Chem. Phys, 16, 8791-8815, 2016.
Trickl, T., Neidl, F., Giehl, H., Perfahl, M., and Vogelmann, H.: Three decades of tropospheric ozone lidar
development at Garmisch-Partenkirchen, Atmos. Meas. Tech., 13, 6357-6390, 2020a.
Trickl, T., Vogelmann, H., Ries, L., and Sprenger, M.: Very high stratospheric influence observed in the free
troposphere over the Northern Alps – just a local phenomenon? Atmos. Chem. Phys., 20, 243-266, 2020b.



Vaisala: Vaisala Radiosonde RS41 Measurement Performance, White Paper, Vaisala, Helsinki (Finland),
https://www.vaisala.com/sites/default/files/documents/WEA-MET-RS41-Performance-White-paper-
B211356EN-B-LOW-v3.pdf, 28 pp. (accessed 7 September 2019), 2017.
Van Malderen, R., Allaart, M. A. F., De Backer, H., Smit, H. G. J., and De Muer, D.: On instrumental errors and
related correction strategies of ozonesondes: possible effect on calculated ozone trends for the nearby sites Uccle
and De Bilt, Atmos. Meas. Tech., 9, 3793–3816, 2016
Viallon, J., Lee, S., Moussay, P., Tworek, K., Peterson, M., and Wielgosz, R. I.: Accurate measurements of
ozone absorption cross-sections in the Hartley band, Atmos. Meas. Tech., 8, 1245-1257, 2015.
Völger, P., Bösenberg, J., and Schult, I.: Scattering Properties of Selected Model Aerosols Calculated at UV-
Wavelengths: Implications for DIAL Measurements of Tropospheric Ozone, Beitr. Phys. Atmosph., 69, 177-

769     187, 1996.

Vömel, H., David, D. E., and Smith, K.: Accuracy of tropospheric and stratospheric water vapor measurements
by the cryogenic frost point hygrometer: Instrumental details and observations, J. Geophys. Res., 112, D08305,
doi: 10.1029/2006JD007224, 14 pp., 2007.
Vömel, H., Naebert, T., Dirksen, R., and Sommer, M.: An update on the uncertainties of water vapor
measurements using Cryogenic Frostpoint Hygrometers, Atmos. Meas. Tech., 9, 3755-3768, 2016.
Vömel, H., Smit, H. G. J., Tarasick, D., Johnson, B., Oltmans, S. J., Selkirk, H., Thompson, A. M., Stauffer, R.
M., Witte, J. C., Davies, J., van Malderen, R., Morris, G. A., Nakano, T., and Stübi, R.: A new method to correct
the electrochemical concentration cell (ECC) ozonesonde time response and its implications for "background
current" and pump efficiency, Atmos. Meas. Tech., 13, 5667–5680, 2020.
Vogelmann, H. and Trickl, T.: Wide-Range Sounding of Free-Tropospheric Water Vapor with a Differential-
Absorption Lidar (DIAL) at a High-Altitude Station, Appl. Opt., 47, 2116-2132, 2008.
Vogelmann, H., Sussmann, R., Trickl, T., and Borsdorff, T.: Intercomparison of atmospheric water vapor
soundings from the differential absorption lidar (DIAL) and the solar FTIR system on Mt. Zugspitze, Atmos.
Meas. Tech., 4, 835-841, 2011.
Vogelmann, H., Sussmann, R., Trickl, T., and Reichardt, A.: Spatiotemporal variability of water vapor
investigated using lidar and FTIR vertical soundings above the Zugspitze, Atmos. Chem. Phys., 14, 3135-3148,

786     2015.

VOTALP II: Vertical Ozone Transport in the Alps II, Final Report for the European Union, Contract Nr.: ENV4
CT970413, Reporting Period 1/3/1998-29/2/2000, H. Kromp-Kolb, Co-ordinator, Universität für Bodenkultur
Wien (Austria), Institut für Meteorologie und Physik, 96 pp., 2000.
Wernli, H., and Davies, H.C.: A Lagrangian-based analysis of extratropical cyclones. I: The method and some
applications. Q.J.R. Meteorol. Soc., 123: 467-489, https://doi.org/10.1002/qj.49712353811, 1997.
Sprenger, M., and Wernli, H.: The LAGRANTO Lagrangian analysis tool – version 2.0, Geosci. Model Dev., 8,
2569–2586, https://doi.org/10.5194/gmd-8-2569-2015, 2015.
Wotava, G., and Kromp-Kolb, H.: The research project VOTALP – general objectives and main results, Atmos.
Environ., 34, 1319-1322, 2000.



Yuan, Y., Ries, L., Petermeier, H., Trickl, T., Leuchner, M., Couret, C., Sohmer, R., Meinhardt, F., and Menzel,
A.: On the diurnal, weekly, and seasonal cycles and annual trends in atmospheric $CO_2$ at Mount Zugspitze,
Germany, during 1981–2016, Atmos. Chem. Phys., 19, 999–1012, https://doi.org/10.5194/acp-19-999-2019,

799    2019.

Zanis, P., Trickl, T., Stohl, A., Wernli, H., Cooper, O., Zerefos, C., Gaeggeler, H., Priller, A., Schnabel, C.,
Scheel, H. E., Kanter, H. J., Tobler, L., Kubik, P. W., Cristofanelli, P., Forster, C., James, P., Gerasopoulos, E.,
Delcloo, A., Papayannis, A., and Claude, H.: Forecast, observation and modelling of a deep stratospheric
intrusion event over Europe, Atmos. Chem. Phys., 3, 763-777, 2003.



**Figures:**

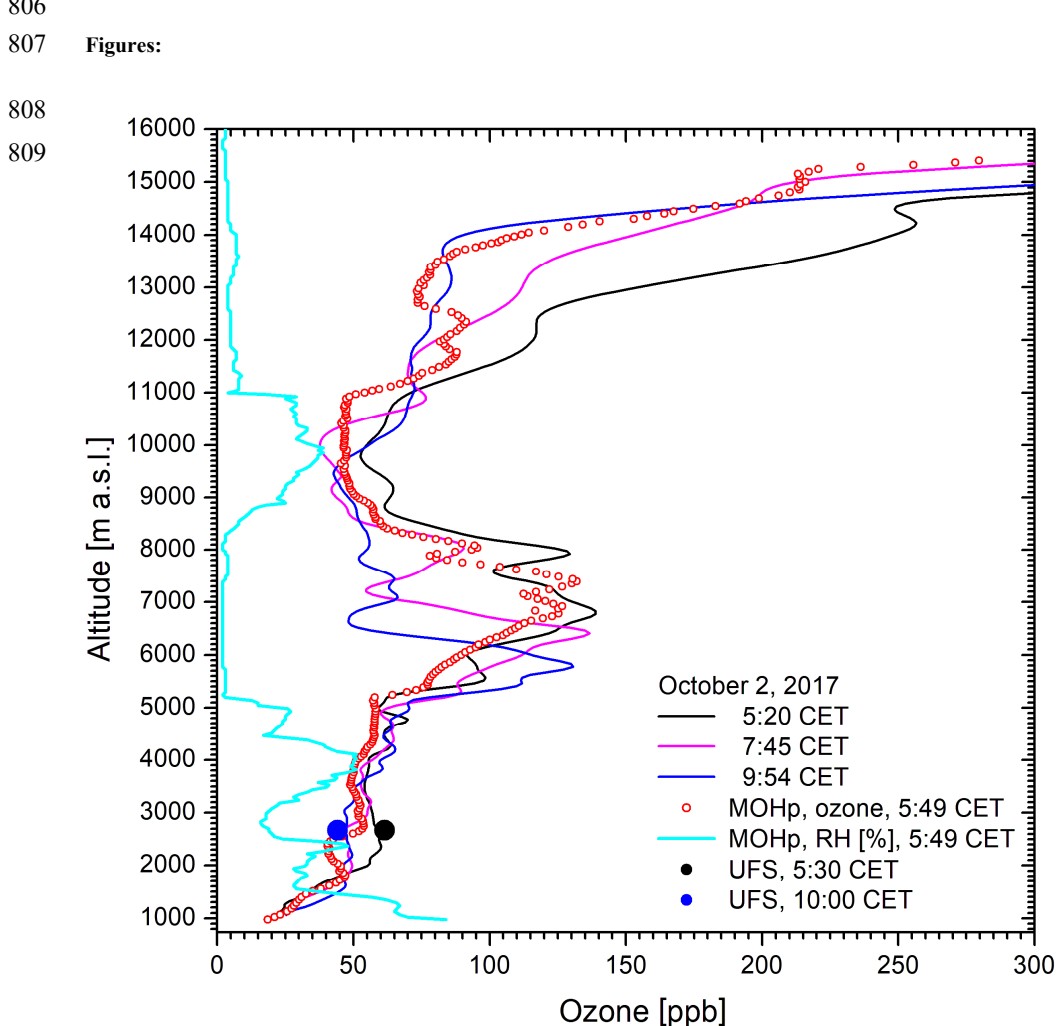

**Fig. 1.** Ozone measurements at Garmisch-Partenkirchen (IFU, UFS) and Hohenpeißenberg (MOHp) on 2
October 2017; the low relative humidity between 5.2 and 8.3 km (RH = 2 %) verifies the presence of a
stratospheric air intrusion. The time for MOHp is the launch time of the sonde.

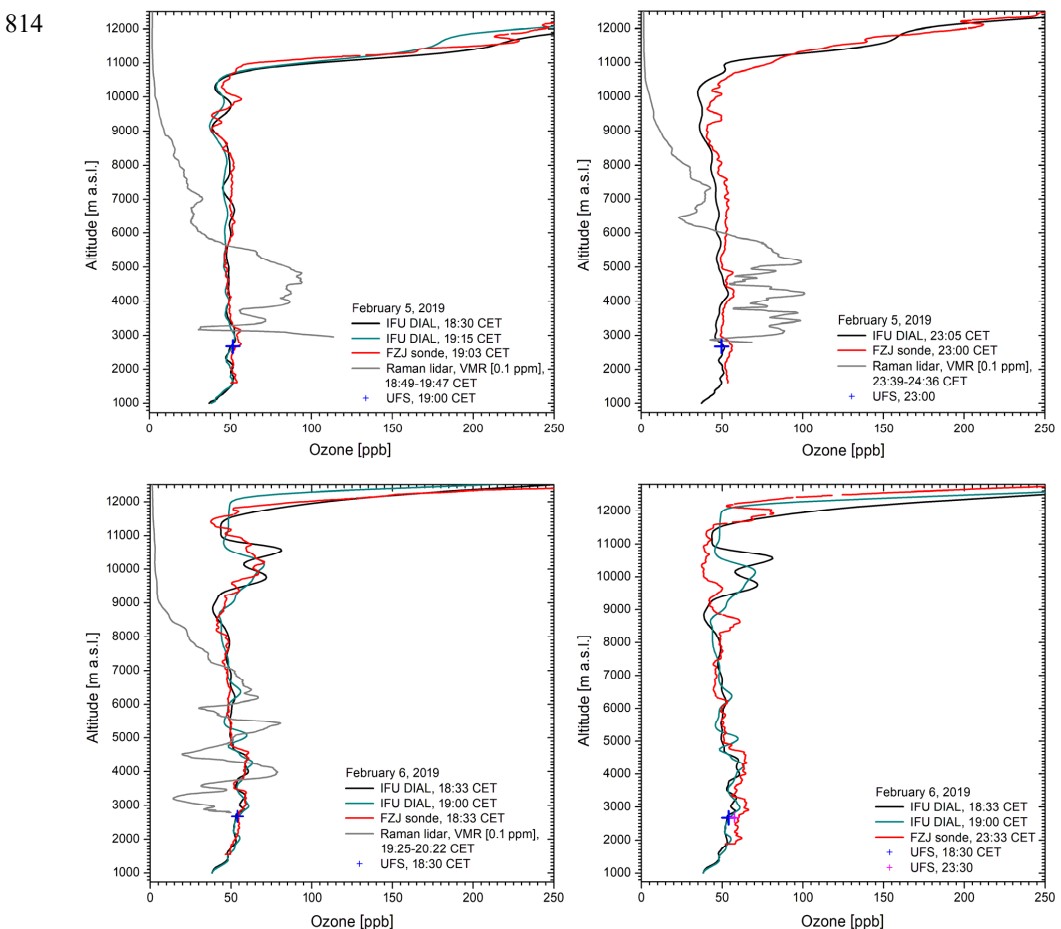

**Fig. 2.** Four ozone measurements on 5 and 6 February 2019 with lidar (IFU), ECC sonde (FZJ) and an *in-situ*
sensor at UFS; for two measurements the FZJ ozone mixing ratios are slightly higher than the lidar results. The
fourth FZJ ozone measurement took place much later than the final lidar measurements which resulted in slightly
larger differences. The lidar results around 10 km on 6 February are uncertain due to a cirrus correction. In order
to visualize more details on the layering we also show water-vapour mixing ratios for roughly co-inciding
measurements of the UFS Raman lidar. The tropospheric structures are strongly smoothed due to the 1-h data-
acquisition time.






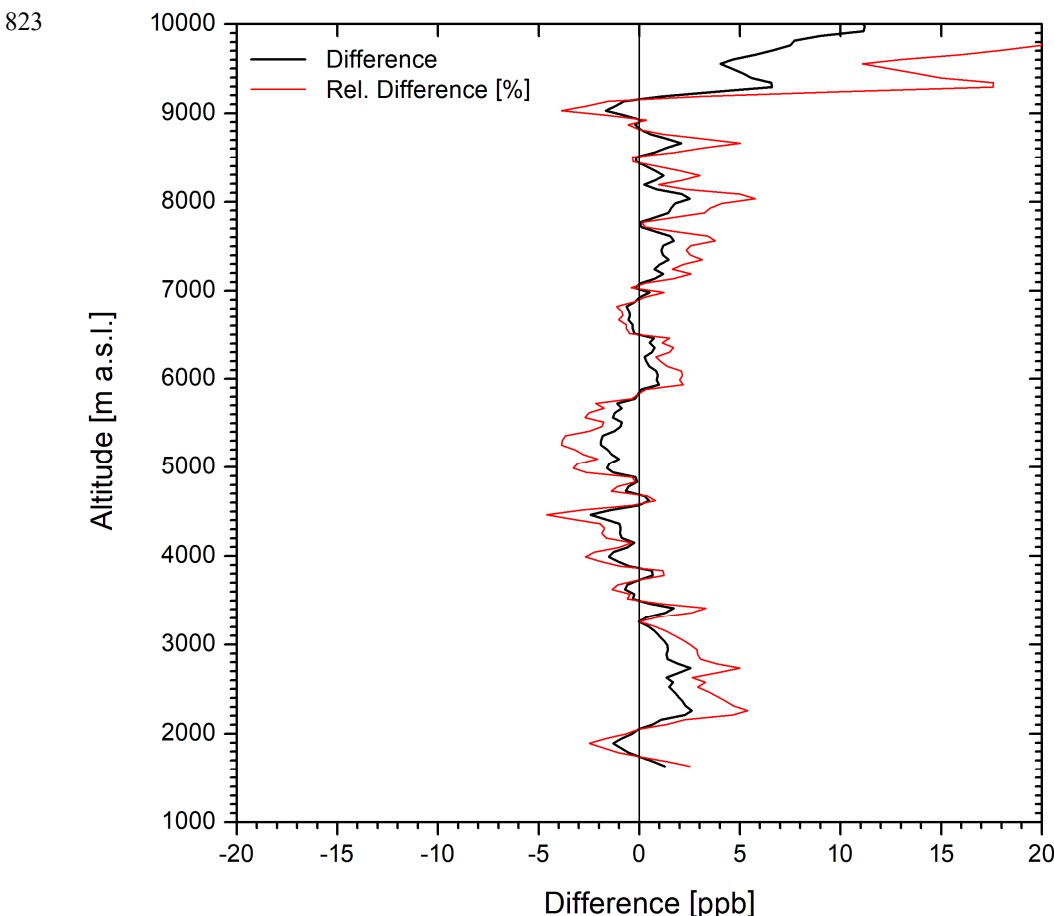

**Fig. 3.** Averaged differences between FZJ ozone sonde and IMK-IFU lidar for the first three comparisons after a

slight offset correction (see text)


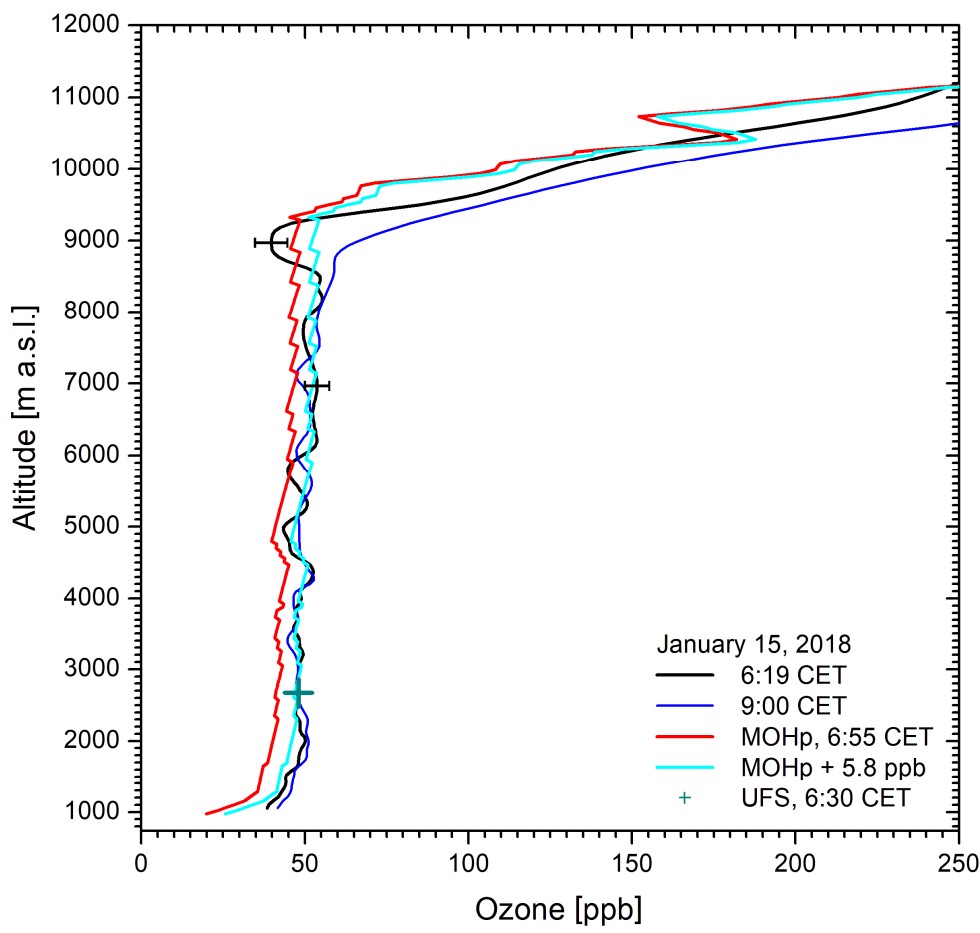

**Fig. 4.** Ozone measurements on 15 January 2018: The MOHp ozone (red) is also shown shifted by 5.8 ppb to match the lidar ozone (cyan), in part the black, in part the blue curve. Differences exist in the tropopause region, which is frequently the case. The sawtooth structure in the MOHp data is due to insufficient digital resolution in the NDACC data base.

834

835

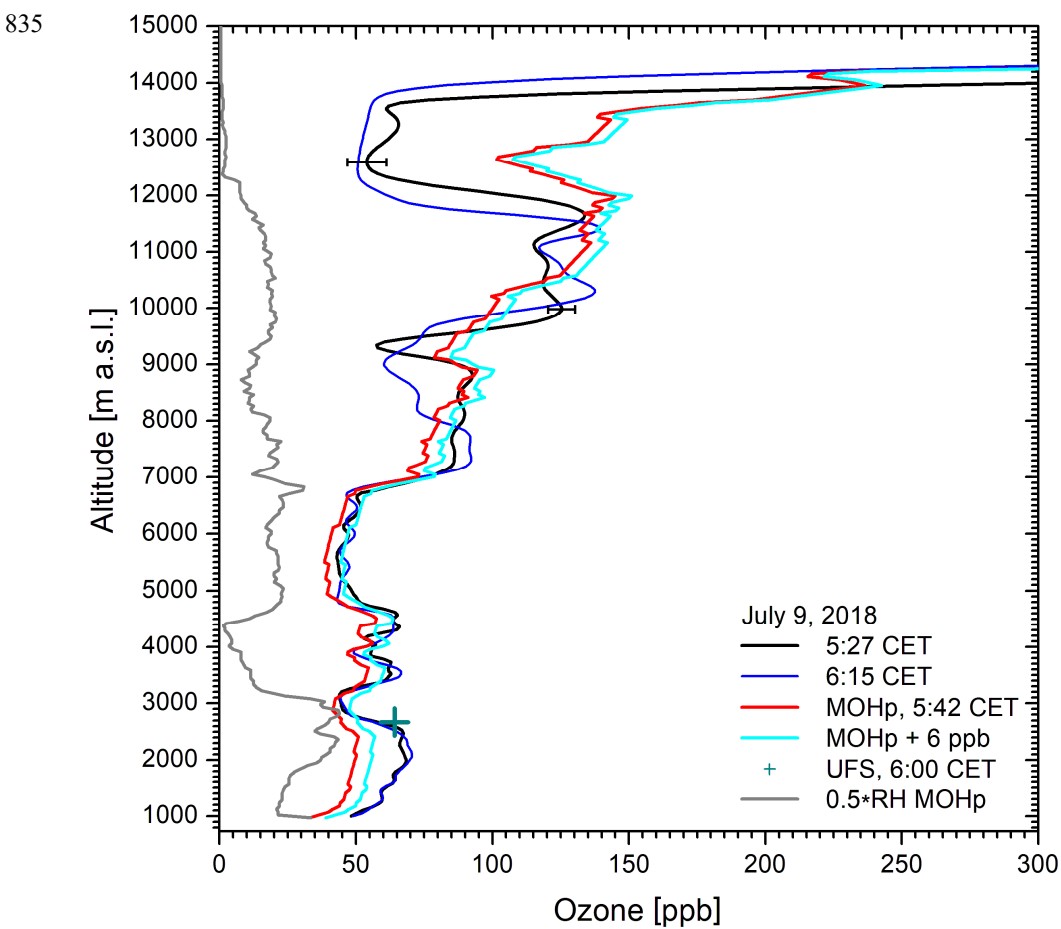

**Fig. 5.** Summertime ozone measurements (July 9, 2018) with pronounced layering; the sonde ozone (red) is
brought to reasonable agreement with the lidar (black curve) above 2.7 km by adding 6 ppb (cyan). Above 9 km
the air masses are no longer comparable. The particularly strong discrepancy of the UFS *in-situ* ozone can be
explained by orographic lifting of the ozone edge at 2.7 km. The moderate RH (grey) in the free troposphere
indicates that the very high ozone values could be due to a stratospheric air component.





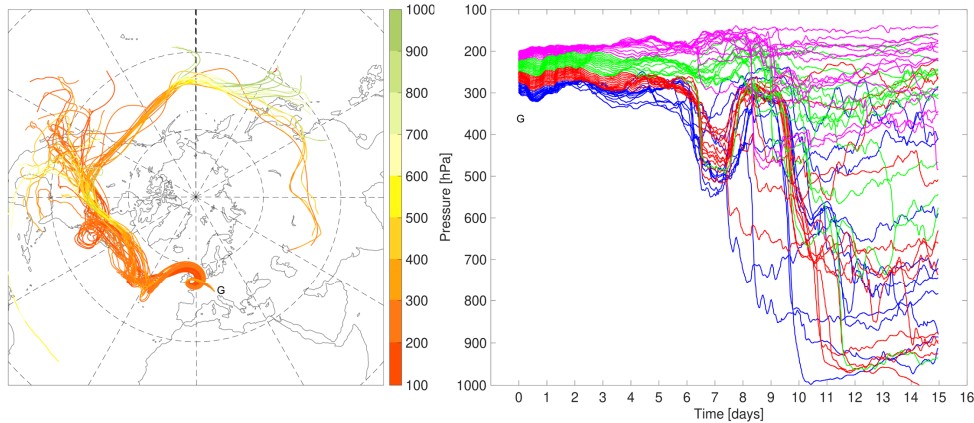

**Fig. 6.** 350-h LAGRANTO backward trajectories, started above Garmisch-Partenkirchen (G) on 9 July 2018 at 7:00 CET

**Fig. 7.** 350-h LAGRANTO backward trajectories, started above Garmisch-Partenkirchen (G) on 9 July 2018 at 8:00 CET


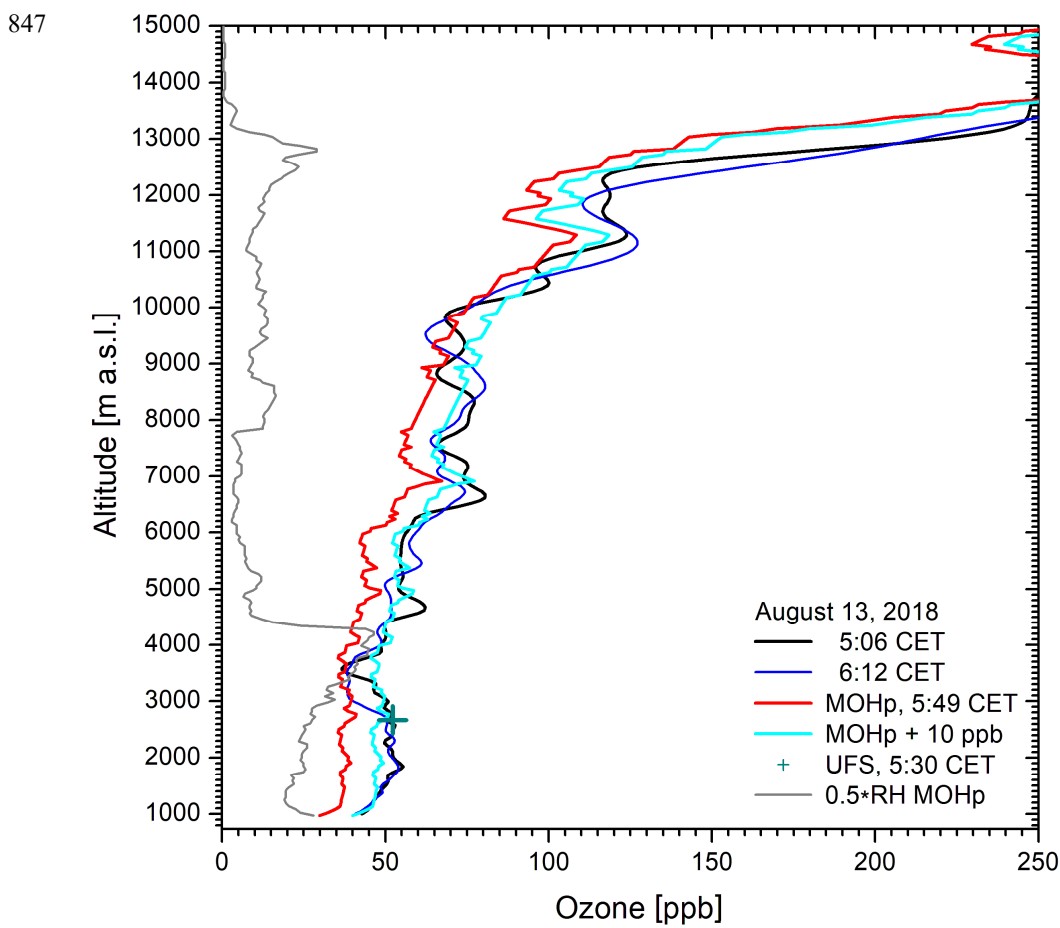

**Fig. 8**. Ozone measurements on 13 August 2018: The agreement of the shifted MOHp ozone profile (cyan) with
the lidar curves is rather good up to 12 km given the high summertime variability. The low to moderate RH
above 4.4 km (grey) indicates that the elevated ozone is partially caused by stratospheric air.




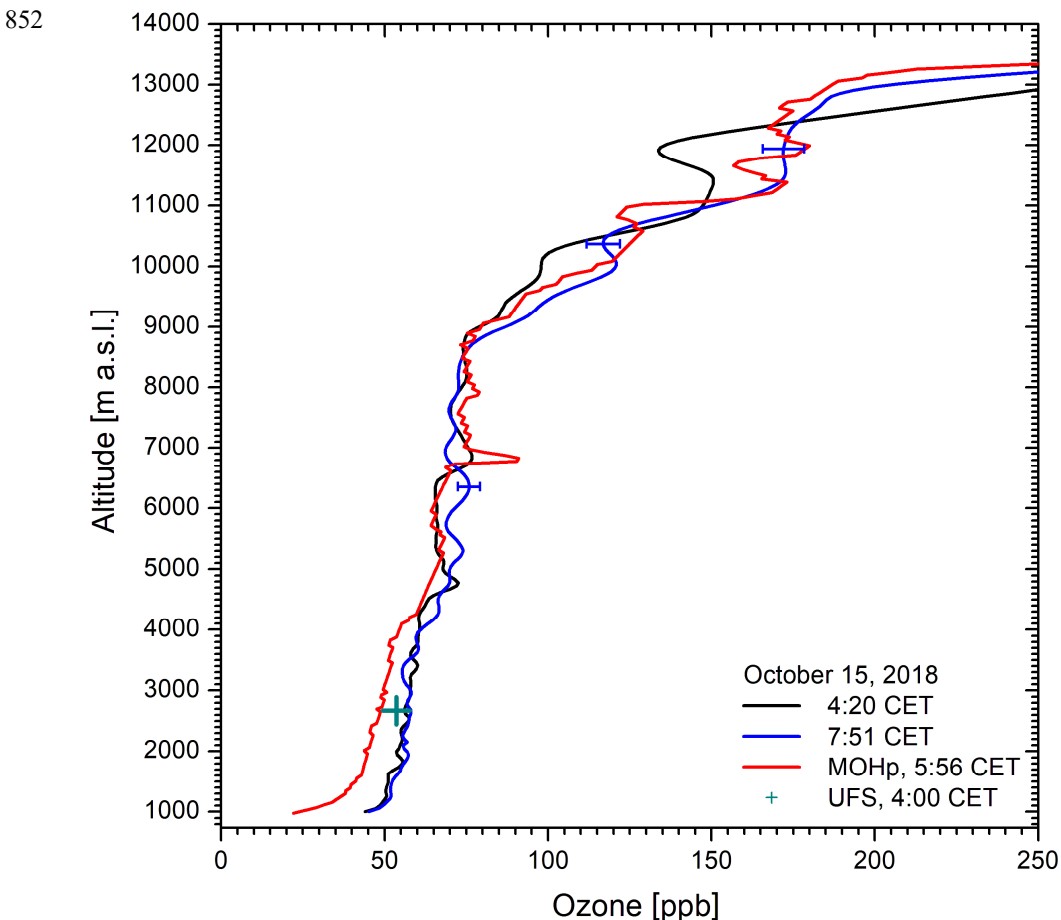

**Fig. 9**. Ozone measurements on 15 October 2018: The MOHp ozone (red) is not shifted. The agreement above
4.3 km is better with the earlier lidar measurement (black), above 7 km better with the blue curve. The lidar data
are strongly smoothed in the stratosphere, as can be seen from the more detailed ozone structure in the sonde
data. This example is one of the two examples with a pronounced low-altitude discrepancy between lidar and
sonde extending to more the 3 km.




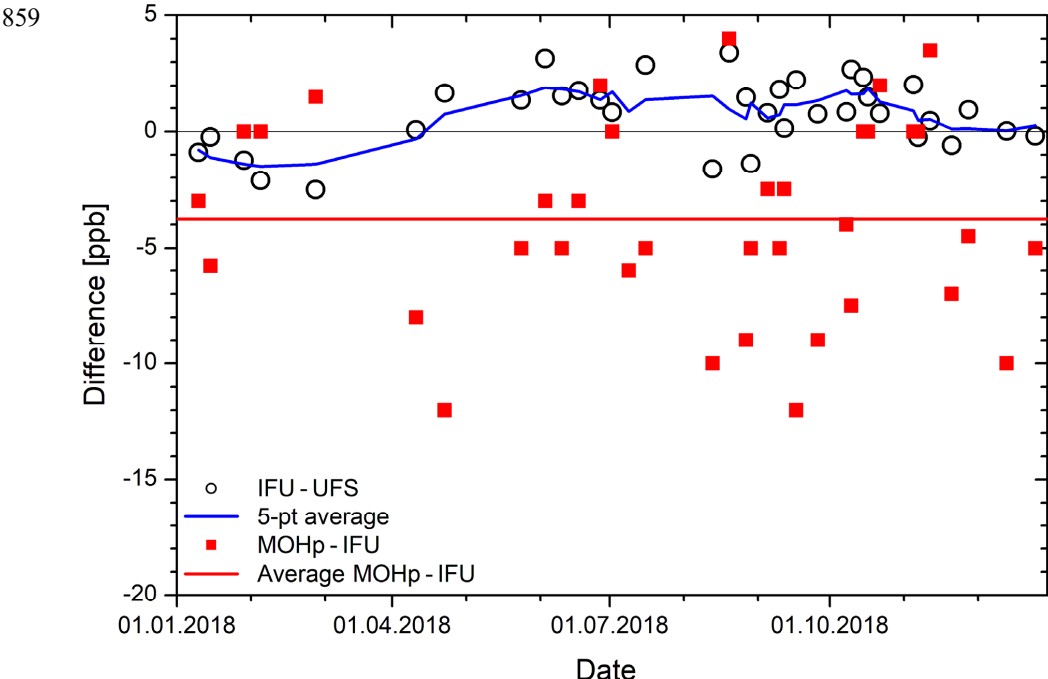

**Fig. 10**. Differences between the ozone values of the IFU DIAL at 2670 m and the UFS routine measurements as
well as the offsets of the MOHp profiles with respect to the DIAL for 35 of the 36 measurement days of the 2018
comparison. The blue curve represents a ±2-point running average of the differences between lidar and station.





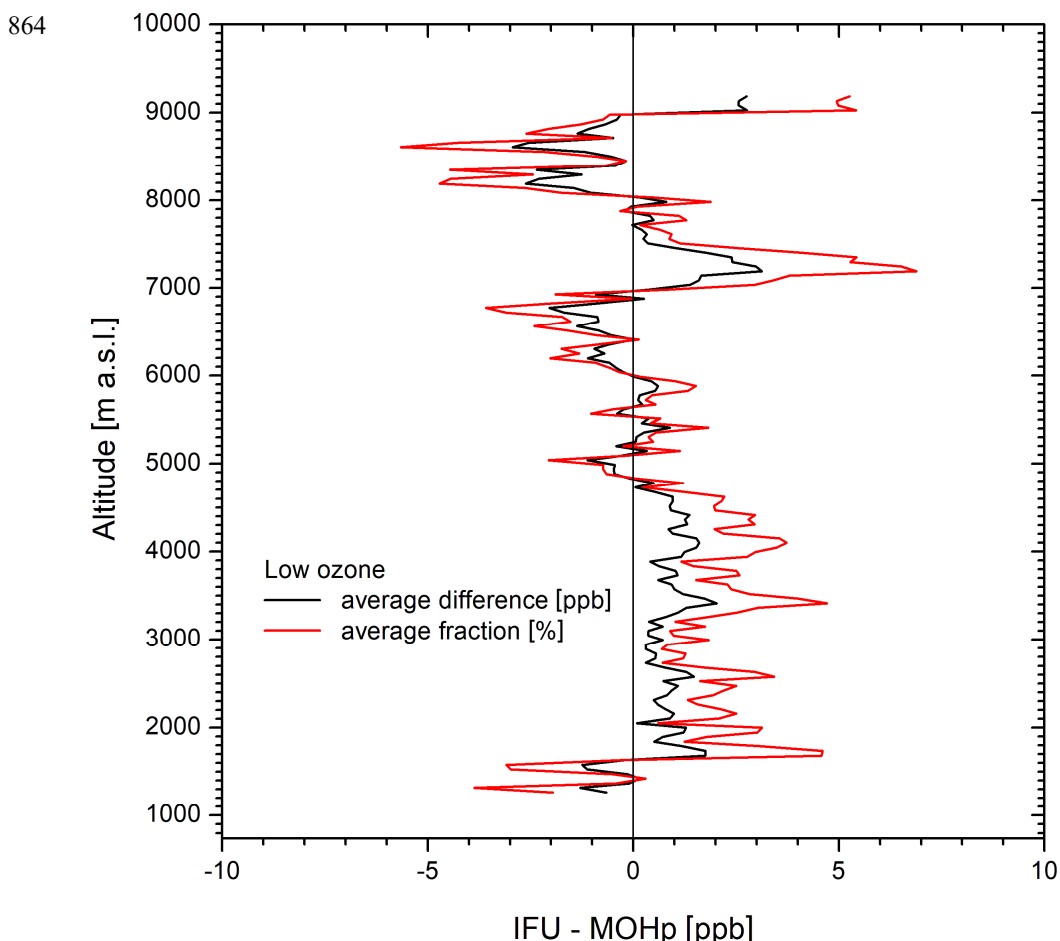

**Fig. 11.** Average differences between IFU lidar and offset-corrected MOHp sonde in 2018 for low-ozone
conditions (based on six cases); the uncertainties may be estimated from the maximum differences around these
altitudes.






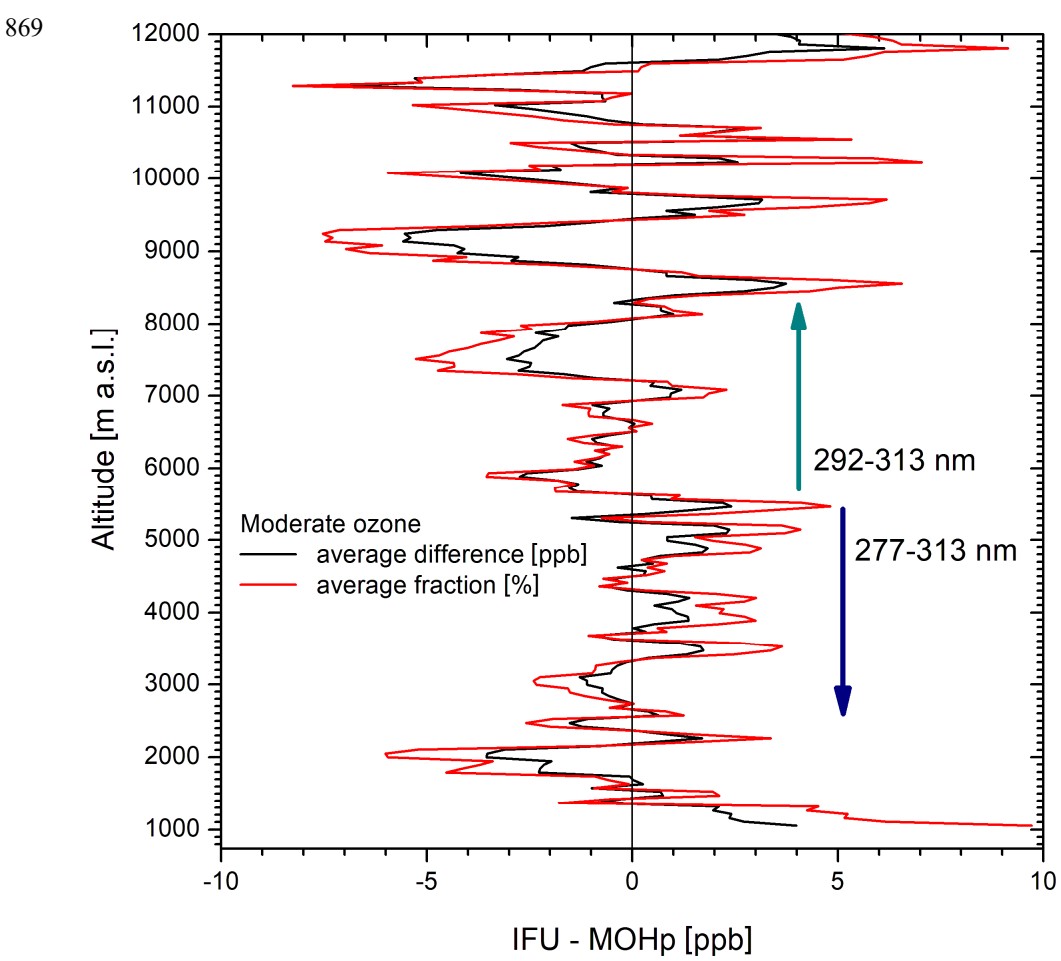

**Fig. 12.** Average differences between IFU lidar and offset-corrected MOHp sonde in 2018 for moderate-ozone
conditions (based on seven cases); we also indicate the altitude ranges of the two wavelength pairs used for the
lidar data evaluation.



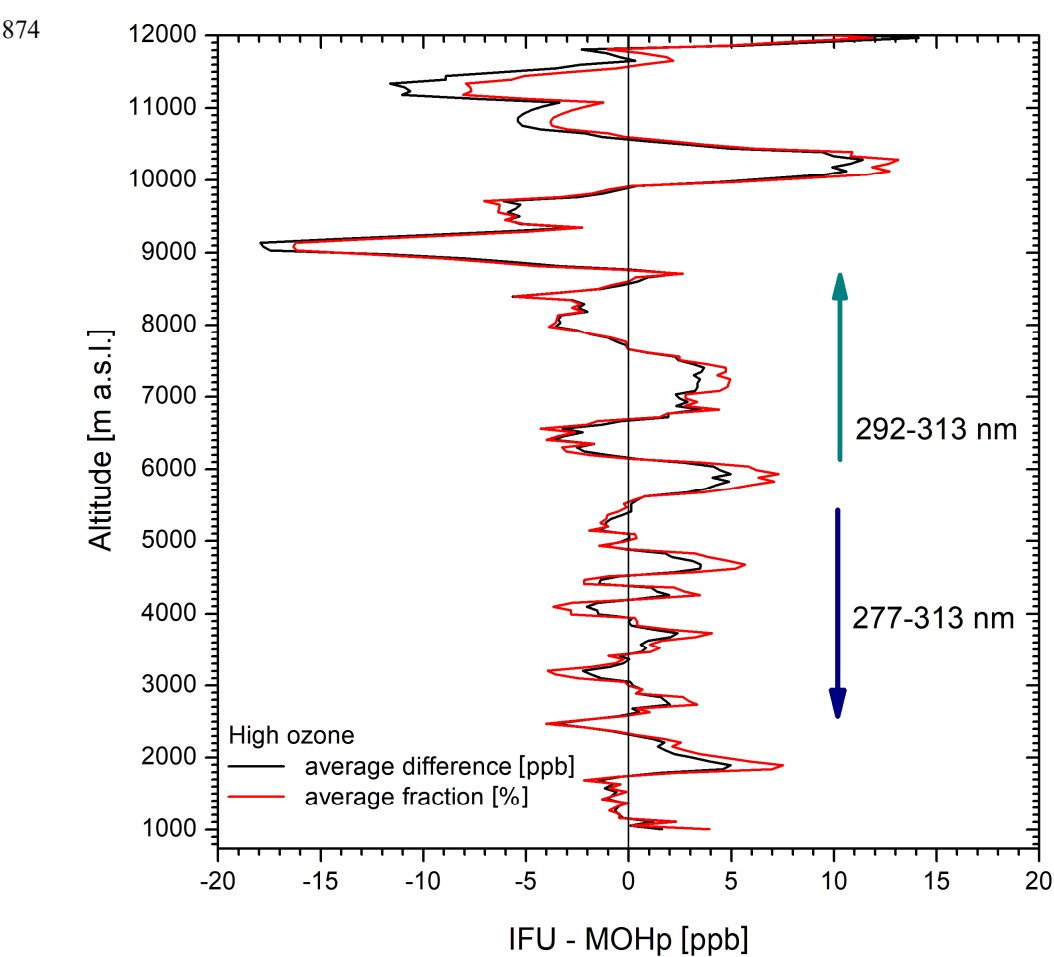


**Fig. 13.** Average differences between IFU lidar and offset-corrected MOHp sonde for high-ozone conditions
(based on six cases); we also indicate the altitude ranges of the two wavelength pairs used for the lidar data
evaluation.







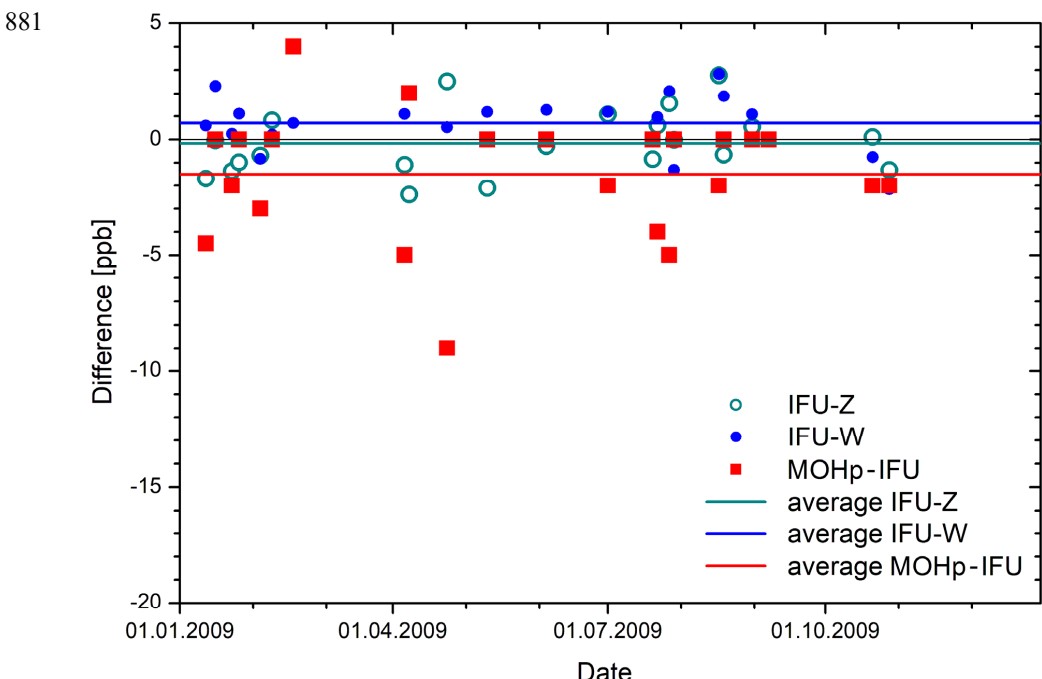

**Fig. 14**. Differences between the ozone mixing ratios of the lidar (IFU) and the stations Zugspitze (Z), Wank
(W) at the summit altitudes, and offsets between lidar and MOHp sonde






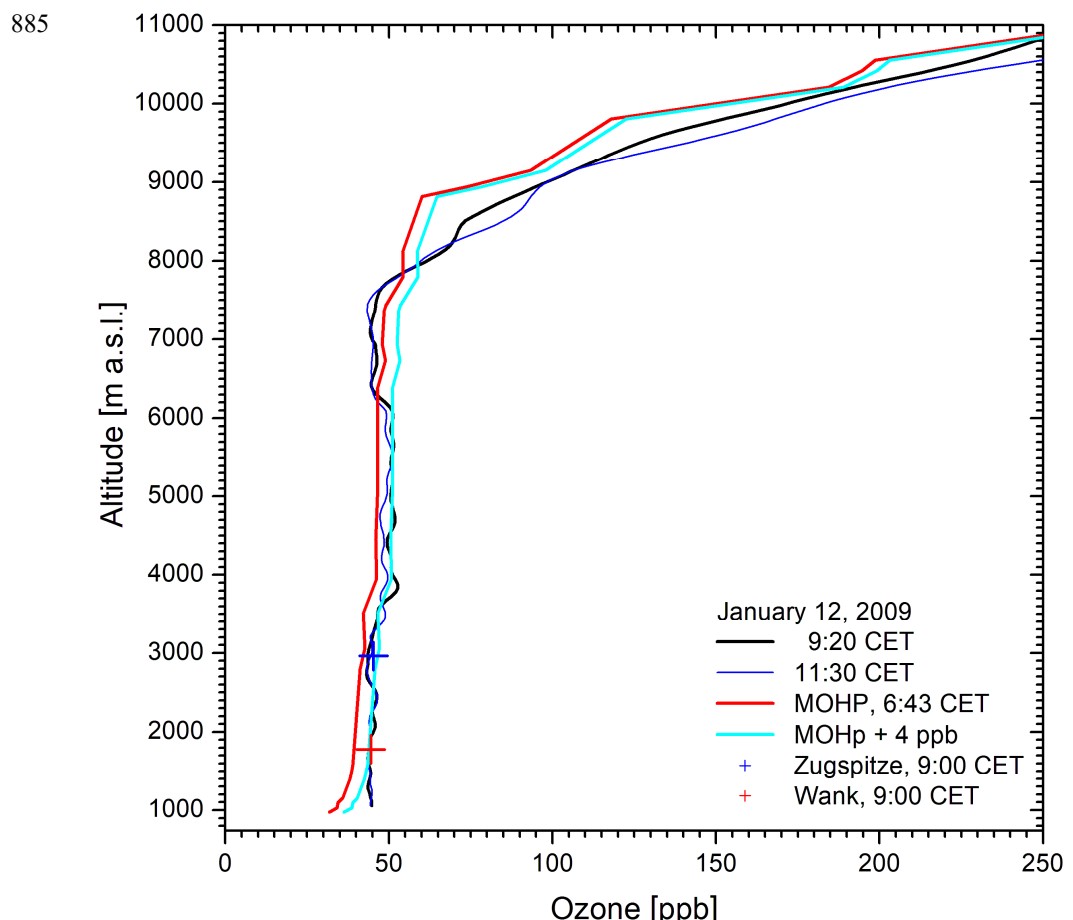

**Fig. 15**. Ozone measurements on 12 January 2009



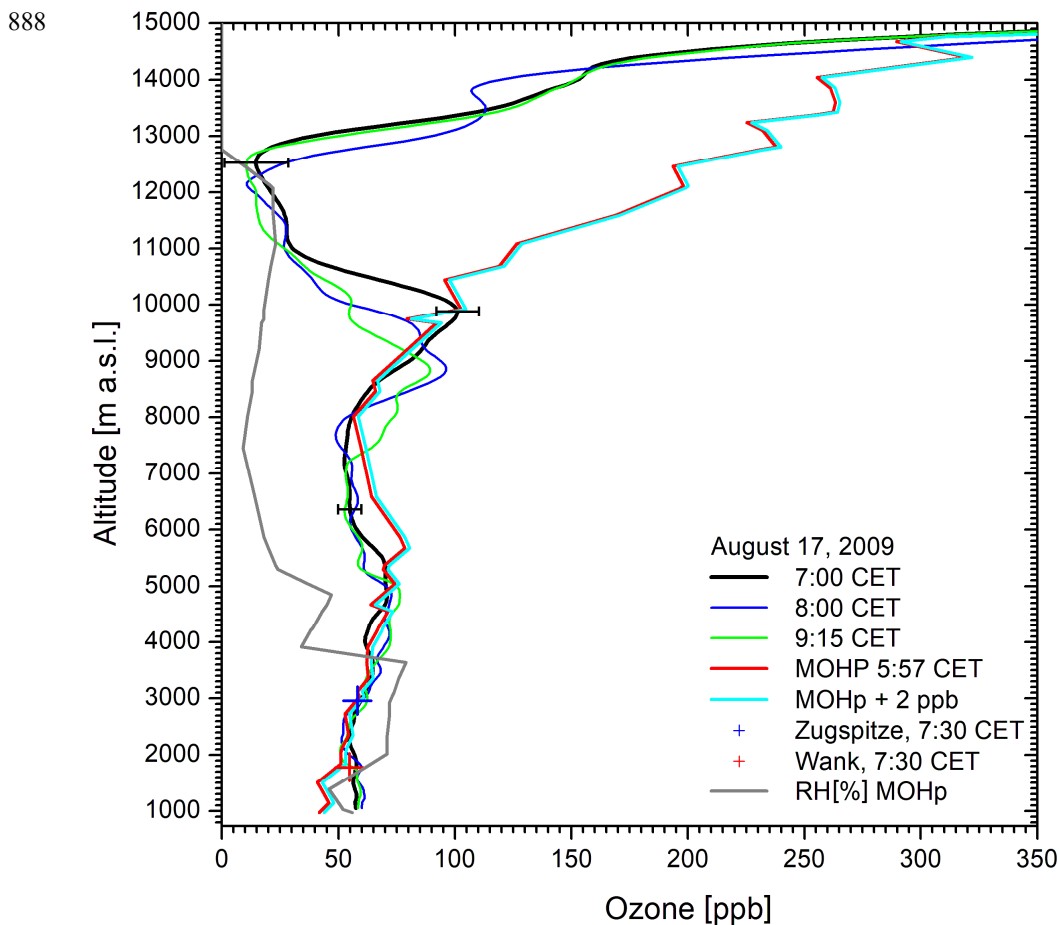

**Fig. 16**. Ozone measurements on 17 August 2009; the structure in the upper troposphere is strongly influenced
by smoothing. The bias between 5.5 and 8 km has not been explained.






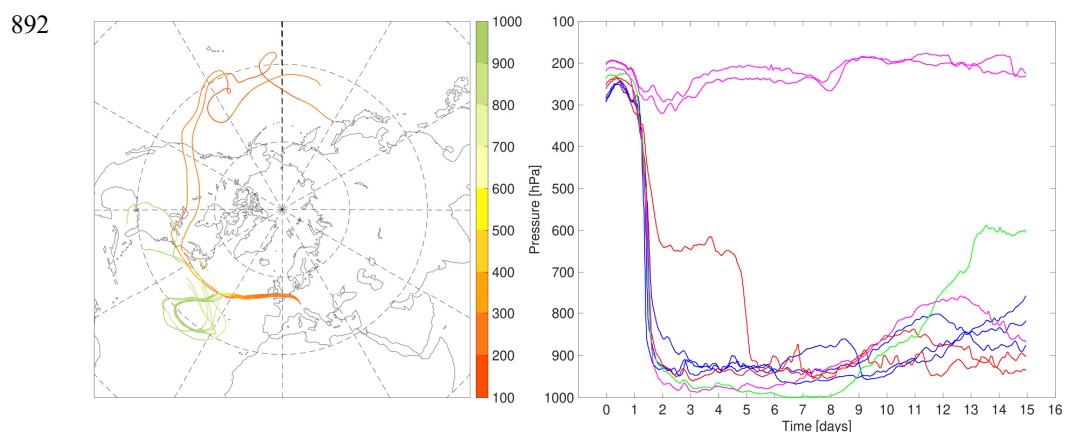

**Fig. 17.** 350-h LAGRANTO backward trajectories, started above Garmisch-Partenkirchen (G) on 9 July 2018 at
7:00 CET



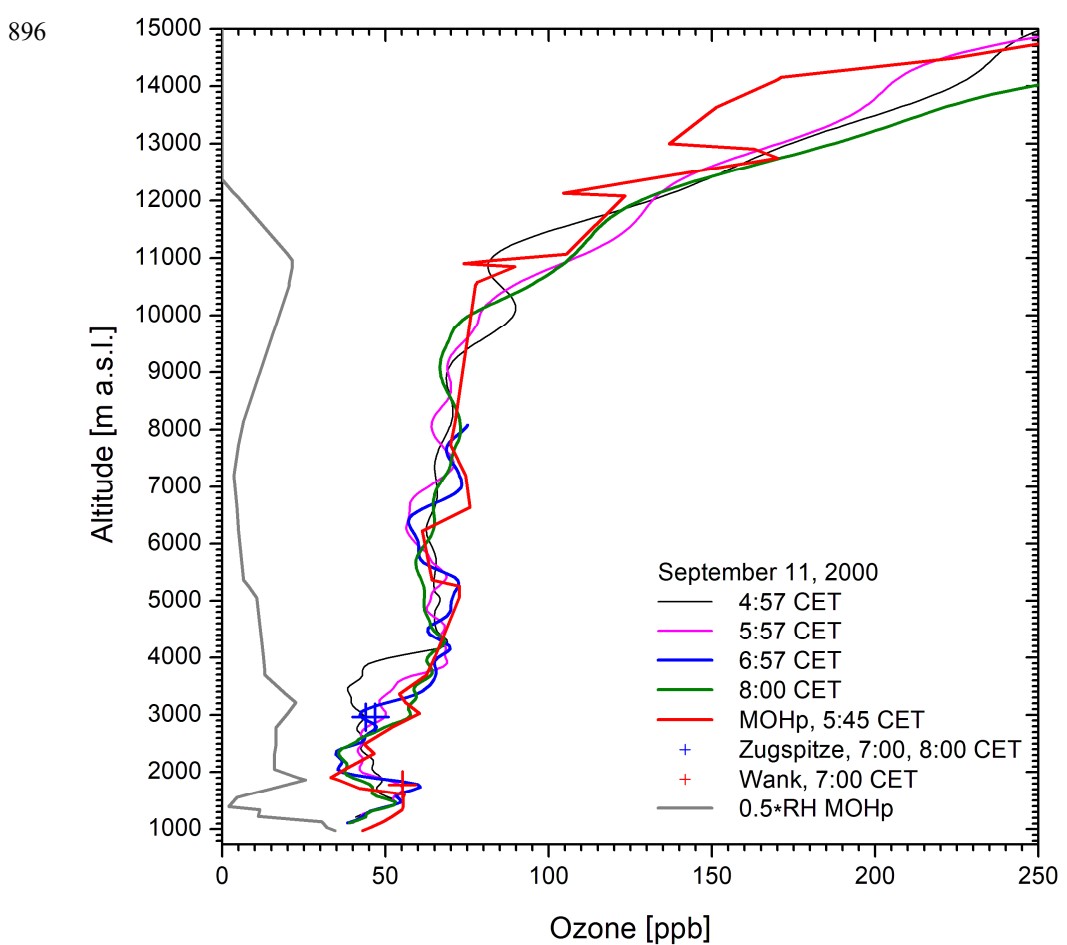

**Fig. 18**. Ozone measurements on 11 September 2000




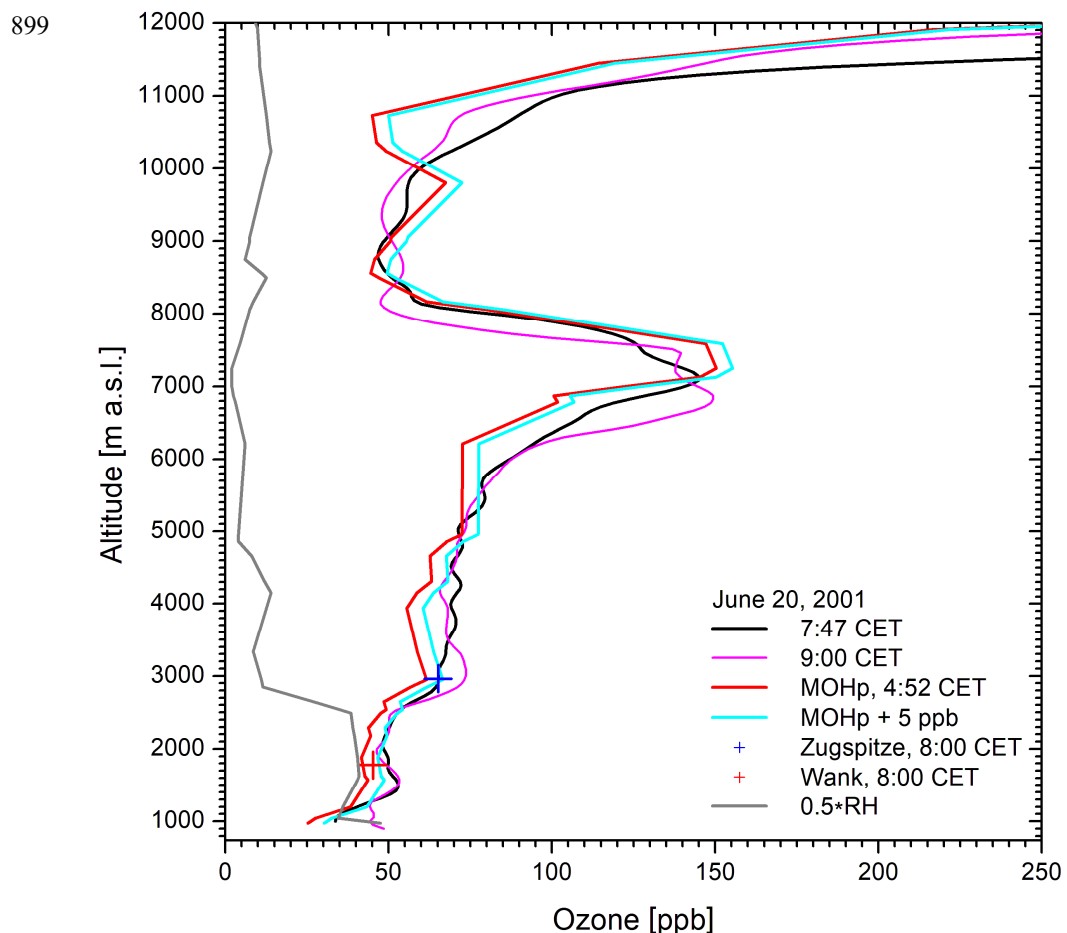

**Fig. 19**. Ozone measurements on 20 June, 2001






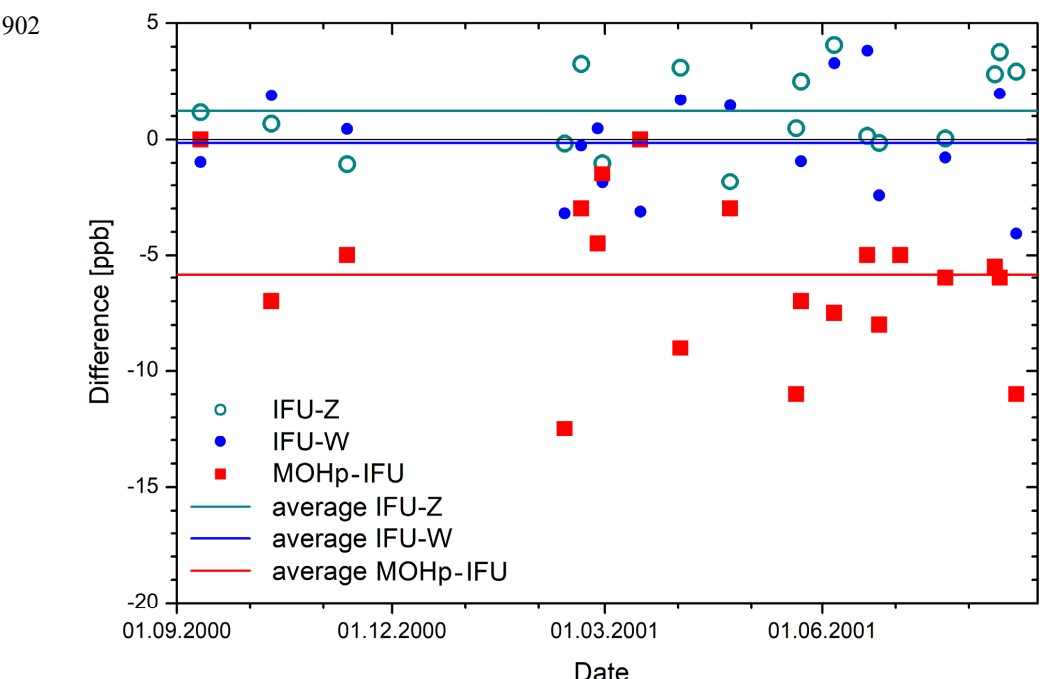

**Fig. 20.** Differences between the ozone mixing ratios of the lidar (IFU) and the stations Zugspitze (Z), Wank
(W) at the summit altitudes, and between lidar and MOHp sonde determined by shifting the sonde profile.

