# Peer review of "Local comparisons of tropospheric ozone: Vertical soundings at two neighbouring stations in Southern Bavaria"

_Atmospheric Measurement Techniques, 2023_

## Referee Comment (RC1)

[revised manuscript text omitted]

---

## Author Response (AR1)

**Reply to the reviewers´ comments on "Local comparisons of tropospheric ozone: Vertical soundings at two neighbouring stations in Southern Bavaria" by Trickl, et al.**

Thomas Trickl, June 4, 2023

We thank the three reviewers for highly favourable, but also helpful critical comments. In our reply we write the reviewers´ comments in italics, our reply in normal text.

**Review 1**

*The paper needs only some minor rather technical corrections (see annotated manuscript).*

All changed but:

Line 18: "DIAL" is not necessary here since it is not used in the rest of the Abstract.

Line 147: Already cited in Line 144!

Lines 174-175: This statement is correct: in the absence of aerosol the only unknown quantities are the atmospheric number density and the ozone density. The atmospheric density can be accurately calculated from the pressure and temperature measurements of nearby radiosonde stations. Thus, ozone remains as the only quantity to be retrieved. I added a reference.

**Review 2**

*Summary:*

*This manuscript compares vertical profiles of ozone observed by the tropospheric DIAL ozone lidar at IMK-IFU Garmisch Partenkirchen, by Brewer-Mast ozonesondes launched at Hohenpeissenberg about 35 km to the northwest, and by ECC ozonesondes launched at the IMK-IFU site. In addition, the manuscript uses continuous surface ozone observations at three high-elevation surface sites about 9 km to the southwest.*

*The manuscript describes some of the technical developments of the lidar system and uses the high-elevation surface observations as anchor points for the vertical profiles. The manuscript then proceeds to describe the comparisons between the different instruments.*

*Comparisons of this type are vital to provide confidence in any of the involved instruments and to be able to characterize the unavoidable biases between them. As such, this manuscript is an important contribution for the understanding of the long term data sets such as the Hohenpeissenberg soundings, but also the IMK-IFU lidar ozone measurements.*

*I would recommend publication after some major revisions, which I detail below.*

After carefully reading the manuscript many times (resulting in a considerable number of corrections) we are impressed about the unbelievable amount of useful comments, and we thank Reviewer 2 for his excellent job!

*Major comments:*

*Some of the comparisons between the lidar and ozonesondes need to be re-written. In multiple figures and descriptions in the text you first apply a bias correction to the ozonesonde data and then state and show an "almost perfect" agreement with the lidar. This is misleading. There is no justification for applying this bias correction other than making the agreement look good. It would be much more interesting to properly describe the bias and show profiles of the actual differences.*

It is obvious that a simple correction formula should preferred since it does not infer additional uncertainty. The side-by-side comparisons with the ECC sondes and also many MOHp Brewer-Mast sondes suggest that a constant offset is a highly reasonable choice. The observed structure of the FZJ profiles agrees unbelievably well with the lidar ozone considering the complexity of the lidar method. Another possibility would be multiplying the profiles by a correction factor. However, the high-ozone situations in summer examples (e.g., Figs. 5 and 8) do not support such an approach. A 10 % correction in the lower troposphere would mean up to twice the shift of the mixing ratio in the upper troposphere. We quickly realized this and, thus, used the constant-offset approach. It would be interesting to study the behaviour of the sondes in detail. This is beyond the scope of this paper.

We do not agree that profiles of the actual differences are the better choice. In this way you cannot exclude that easily layers where obviously atmospheric transport causes differences! For 2009 the agreement is excellent just up to 6 km (unfortunately, there are just eight comparisons with good temporal proximity). Apart from atmospheric differences we cannot exclude technical and mathematical issues above this altitude, caused by the weaker absorption at 292 nm and possibly sub-optimum baseline corrections within the noisy signals. Here, a revision including the more advanced experience from the final years is now planned.

*You show a statistics of the bias (Figures 10, 14, and 20), which is essential for this paper. However, it would be good to understand the justification for a constant bias, i.e. it would be better to show differences of the profiles. In addition, you should discuss over which altitude range the average difference is determined. Figures 11-13 are outright misleading in this context. I provide more suggestions in the detailed comments.*

The justification is now given. The variability of the mixing ratios is too high for a more sophisticated approach. The constant correction allows us to estimate uncertainties of the lidar. Although the uncertainties are most likely overestimated due to a residual atmospheric component the result is satisfactory and there is no need to apply a better correction formula. The altitude range used for the offset determination is explained. We focus on the range covered by the highly accurate mixing ratios derived from the 277-313 nm wavelength pair. This range ends at about 6 km above the ground. However, in many cases good agreement is found up to the upper troposphere.

*Throughout the manuscript it would be good if it was clear what is considered as the reference, i.e. the less biased observation. I assume it is the high-elevation surface observations first, then the lidar and the ECC ozone sondes, and last the Brewer-Mast sondes. This is nowhere explicitly described and may not be easy to do. However, it would be useful to point out which instrument is suspected of having a bias. This could be done in the discussion section.*

We are surprised by this comment since this was planned. We tried to improve this.

*In a few cases there appear to be selective exclusions of profiles without good instrumental justification. I give some examples below.*

We do not exclude profiles. In a single case we exclude a comparison between lidar and UFS in which a concentration edge exists at 2.7 km that causes a difference most likely because of orographic lifting above the mountain, a well understood effect.

**Detailed comments:**

*In the abstract it would be useful to give the distance between IMK_IFU and Hohenpeissenberg and Zugspitze.*

The distance of 38 km has been present in line 27! We added the maximum distance of the in-situ stations (< 9 km).

*Line 29: Here a bias and an uncertainty is provided. This should be expanded throughout the manuscript. What does the uncertainty refer to? You use the term "full error", which is not well defined. Is it a single standard deviation? Is it the result of a larger analysis including additional factors?*

"Full error" is the maximum deviation found, i.e., not just a statistical estimate. We now write "maximum of deviations".

*Line 24: It would be good to give a relative offset in addition to the absolute value.*

Improved!

*Line 30: Here is a rare indication that the sonde data are biased. It is not clear how they could be recalibrated with station data. Which station? IMK-IFU? Wank? Zugspitze? Schneefernerhaus?*

This depends on the year. Until 2011 the Zugspitze ozone is the best choice since this is the highest-lying station with rare influence from the boundary layer. Afterwards, the is no other choice than UFS. We added this information.

*Some other measurements at Hohenpeissenberg? Recalibration based on other data may remove instrumental bias, but will not remove co-location and timing errors. If co-location mismatch is the significant, then recalibration may not reduce the variability in the comparison and only reduce a mean bias.*

As I learnt from our co-authors the MOHp data are (unfortunately) just optimized by comparison with the parallel Dobson measurements (column). Unfortunately, no ground calibration has been made. Indeed. a residual uncertainty will remain since not all measurements can be perfectly corrected. However, it could be better than a just mean correction. A mean correction is the only approach before the Zugspitze measurements in 1978.

*Line 63: "at short intervals": What is the temporal resolution that the lidar observations can achieve? I assume you meant to write "with a temporal resolution of less then xxx minutes"; however, I don't know what xxx would be. This should be expanded on in the instrument description.*

Added!

*Line 90: What are the criteria for air-mass matching? This should be described. It might be useful to show how sensitive the comparisons are to these criteria.*

We explicitly studied this for water vapour (see line 87 for two references). As to line 90: We assumed sufficient air-ass matching for the result of the comparison. This is a speculation, and we slightly modified the sentence.

*Lines 94-95: Do you have a reference for this statement? It would be very reasonable to expect a difference in the boundary layer ozone concentrations in particular comparing the mountaintop location of Hohenpeissenberg with the valley location at Garmisch Partenkirchen, which are roughly at the same elevation.*

I cannot remember where I read this statement. Maybe it was mentioned during a conversation many years ago. The difference becomes obvious from the examples shown in this manuscript anyway. The suspected differences were tentatively ascribed to a closer proximity of MOHp to

urban areas. An altitude of MOHp (1 km) is below the height of the boundary layer. However, it is difficult to understand why such a difference should exist at 3 km.

*Line 113: The background current is usually given in uA, not hPa. Is the limit a background current, or an ozone partial pressure?*

Indeed, in the ozone sonde community, background currents are usually given in μA. However, for people outside of the ozone sonde community, μA is not a meaningful quantity. Therefore, we have given the upper limit for the allowed background (≈0.07 μA) in ppb. We now add "corresponding to" in the revised text. The background of most sondes launched is well below this threshold.

*Line 114: The assumption of a constant pump temperature is obviously a weak assumption. Could that explain some of the biases described later?*

On the contrary, the assumption of a constant pump temperature is a pretty good assumption throughout the troposphere. Typically, Brewer-Mast pump temperatures at Hohenpeißenberg decrease from 296 K at launch to 291 K at 10 km (and 276 K at 30 km). As mentioned in the text, assuming constant 300 K leads to a slight overestimation (which partly compensates the insufficient pump correction, e.g. by adding 9% at 30 km), but in the troposphere the overestimation is small and is almost constant: 1.3% at the ground and 3% at 10 km. Assuming 50 ppb of ozone in the troposphere, the corresponding overestimation is 0.65 ppb and 1.5 ppb, respectively.

This is small compared with the range of biases found in the paper, which is about 4 ± 4 ppb low bias for the MOHp ozone sondes.

*Lines 115f: If I understand it correctly, a time response correction may be significant at low ozone concentrations such as in the troposphere even without steep gradients. However, I don't know if anybody has looked into time response issues of Brewer-Mast sondes.*

The response time of Brewer-Mast sondes has been looked into in the past (e.g., De Muer and Malcorps, 1984). It is essentially the same as for ECC sondes, about 20 s (see, e.g., Vömel et al., 2020).

Current practice for both ECC and Brewer-Mast sonde types is to not correct for the response time. The altitude shift is just of the order of 0.1 km. Thus, an influence of this shift is just seen in the presence of strong ozone gradients. The main regions, where a response time correction makes a difference, are pronounced stratospheric intrusions or the lowermost stratosphere, where ozone increases dramatically, both in partial pressure and in mixing ratio.

*Lines 116ff: Could you give a range for typical correction factors for the Brewer-Mast sondes. Could their tropospheric effect contribute to the biases seen? I assume this is not done for the ECC measurements by the FZJ. This should be explicitly stated in section 2.2.*

The average correction factor is 1.08 (added). The standard deviation (1 sigma) of the individual sonde correction factors is 0.05. This means that, on average, the Dobson correction adds 8% (or about 4 to 8 ppbV) to the tropospheric values, and more to the stratospheric values.

In summary, all the (minor) issues related the reviewer's comments on lines 113 to 116 mean that the Brewer-Mast sonde data as used here should slightly over-estimate tropospheric ozone - very much in contrast to the slightly negative bias indicated by the comparison with the lidar and ground-station data in the paper. So, the main conclusion of the paper, that the Hohenpeißenberg ozone sonde data underestimate tropospheric ozone by 4±4 ppbV remains valid. This 2 to 10 % underestimation of tropospheric ozone by Brewer-Mast sondes is in general agreement with several

(but not all) previous intercomparisons, as, e.g., summarized by Deshler et al. (2008, Atmospheric comparison of electrochemical cell ozonesondes from different manufacturers, and with different cathode solution strengths: The Balloon Experiment on Standards for Ozonesondes, J. Geophys. Res., 113, D04307, doi:10.1029/2007JD008975).

*Line 137ff: The Vaisala RS41SGM has the ability to buffer data. In future campaigns, this could be used to span the altitude region of missing telemetry coverage.*

Thank you for this very helpful hint. At the time of the campaign we were not aware about the RS41-SGM radiosonde and their capabilities. We will definitely consider the usage of the RS41-SGM radiosonde in future campaigns with this kind of receiver and launch site set-up.

*Line 151: Is the data acquisition time of 41 s the minimum temporal resolution for profiling?*

No! It is the maximum time possible to take data without transfer to the computer. We added "for the maximum number of 4096 laser shots". 4096 shots turned out to be sufficient except for high-ozone situations. Electronic circuits were developed for stopping the trigger pulses during readout periods, but the installation was postponed because of temporary laser issues prohibiting longer unattended operation of the laser.

*Line 157ff: It would be good if you could show a profile of the expected uncertainty of the DIAL system.*

In numerous publications we inserted error bars, but just in the upper troposphere, because they are rather low for the first kilometres. We thought to add a copy of Table 4 of Trickl et al. (2020a), but this table is perhaps too explicit since it contains also the specifications for different development states of the DIAL. We now cite that table and insert the uncertainties for the final operation period of the system.

*Line 162: What is "an internal quality control"? Can you elaborate?*

We add "By comparing the resulting ozone profiles".

*Line 181f: Please move this sentence to line 151 and elaborate.*

Changed!

*Line 182f: If this option wasn't used, then why mention it at all?*

Good point! However, this option is called for in the Discussion.

*Line 189: You refer to the surface observations as in situ measurements. Ozonesonde observations are also in situ observations, which causes some confusion. I would suggest referring to the surface observations as "high elevation surface observations" instead of "in situ observations".*

Adopted!

*Throughout: The English guidelines by AMT ask not to italicize "in situ".*

I know this from long experience with ACP, and, despite my personal tradition, I made these changes.

*Lines 252f: This is one example of many where you state that there is "outstanding agreement provided that a small constant offset is applied". Either you apply an offset correction or the agreement is outstanding, but not both. Instead, I would suggest to modify all figures showing profiles such that only the uncorrected profiles are shown and in the same figure a panel to the right a profile of the difference(s) is shown. This would show honestly how the offset varies with*

*altitude. In addition, this panel could indicate the altitude range over which the constant offset is calculated. The difference plot could also include the profwe dile of the lidar uncertainty, which gives a reference for what bias to expect introduced by the lidar.*

We carefully examined the text and made adjustments where necessary. In line 252 the description looks clear to us. The average variations of the three profiles coinciding in time are shown in Fig. 3 anyway. Given this performance we do not see any need for justification!

*Lines 253f: Why are the DIAL profiles smoothed? Can you show the effect of the smoothing? Isn't the temporal averaging sufficient to smooth out the profile?*

We realized that this information is, indeed, missing in Sec. 2.4: Added! The raw data are stored at 7.5-m intervals. Since the retrieval is based on calculating the slopes of raw data the data noise is amplified. Thus, the ozone profiles must be smoothed. It is unrealistic to expect reasonable results on a 7.5-m grid.

*Lines 274-280: This is another example, where the offset handling is not appropriate. It would be good to see profiles of the offsets and in addition a statistics of the variability of the offsets across all profiles. I believe this is done in Figure 10, but since it is mentioned here, this would need to be moved up.*

As mentioned the description is now improved! Indeed, Fig. 10 shows the statistics of the offsets. The numbers had already been included in the text.

*Line 302: Another example of the offset handling. If you have to add 5.8 ppbv, the profiles do not match.*

Changed!

*Lines 316ff: Throughout the manuscript, I am missing a discussion of the wind direction. If the sonde launched at MOHP flies straight South, the matching should get better. If it flies straight East, then matching may be more challenging.*

This is done here by trajectory analysis! However, the wind speed differs at different altitudes which, still, makes a quantitative analysis difficult. The sonde could be even south of us after a rise of 0.5 h. We added some text to the HYSPLIT paragraph.

*Lines 333ff: To what altitude does this statement refer? The sonde RH is very low throughout the troposphere, i.e. descending motion is likely. Clearly identifying stratospheric intrusions is a little trickier under these conditions.*

The altitude is now added. We usually use trajectories in addition. Here, we see a four-day descent even for Garmisch-Partenkirchen. Thus, we hesitate to go into more detail.

*Line 339: How could you support that the differences are "mostly not related to differences in air-composition"? This is not obvious.*

This is correct, thank you! Changed!

*Lines 348ff: Before discarding this particular profile, it would be essential to show the horizontal distribution of the ozone distribution based on satellite or modeling output. This Figure should include the track of the ozone sonde. All profiles will have co-location challenges and this case may just be a slightly more challenging.*

We think that such an analysis would suffer from uncertainty in the satellite or modelling vertical resolution. The lidar performance excludes any doubt. The existence of a concentration step at this altitude is confirmed in the MOHp profile.

*Line 353: Is a bias correction due to cross section corrections by 1.8% statistically significant?*

As shown in the statistics of the comparisons the uncertainty of the ozone data derived from the lidar measurements is higher than this. Nevertheless, it is interesting not to neglect this calibration offset. As is shown for all three years and all stations there seems to be a negative bias of the station data of 0.6 ppb ± 0.6 ppb which almost coincides with the known bias of the WMO calibration and which is now included in the abstract and in Sec. 4. Given the technical issues of the lidar technique we are very happy with the low "ozone noise" derived. Also the orographic issues of station measurements could have easily led to a less favourable result!

*Lines 356ff: I find the apparent seasonal cycle not convincing. Given the length of the total observational record, there should be many more data that allow comparing the lidar measurements with the high-elevation surface observations. If it can be shown there, then the argument would be much stronger. As is, the number of data points that indicate an apparent seasonal cycle is quite small. In addition, there may have been some selection of profiles that are causing this seasonal cycle.*

We removed the discussion of the seasonal cycle.

*Line 375ff: This is another example of selective matching. The profiles could be excluded because the co-location is too bad, not because the profiles don't match. That co-location threshold should then be applied to all profiles.*

The number of these layers is quite limited. Thus, the altitude ranges accepted in the comparison clearly dominate! Our purpose is to avoid to infer too much of the atmospheric variability. This is the advantage of the offset correction method.

*Line 378ff: Is there any reason to believe that instrumental issues in either instrument should cause such an oscillation in the bias between the instruments? There is obviously a strong reason to believe that spatiotemporal matching causes oscillations of that sort.*

Lidar sounding is technically extremely delicate requiring highly linear signal processing over eight decades. The more years of experience we have covered the more confident we have become. These findings are helpful indeed!

*Line 381: Figure 13 does not support that statement. This is acknowledged in line 382, i.e. this statement appears to be unsupported and incorrect. Please remove or justify.*

In principle, this is what one should expect because the 277-nm-313-nm ozone data become noisy in that altitude range (which is the reason to use the other wavelength pair). However, just the central panel (old Fig. 12) shows this effect clearly. In order to avoid complex explanations we removed the two sentences.

*Lines 386ff: This is another egregious misstatement. The 2018 analysis does reveal a significant bias, which has been removed!*

Corrected! Thank you!!!!!

*Lines 403ff: Not just temporal proximity is important, the spatial proximity needs to be considered as well.*

The spatial distance is always the same and could not be opimized!!!

*Lines 414ff: I believe this is the first description how the offset correction has been derived. This description needs to come before any biases are discussed. Ideally, the same metric is used for all profiles, i.e. the bias should be calculated over the same altitude range for all soundings. If that has been done, please say so.*

We agree and add the necessary statements.

*Lines 426ff: The RH is still quite low and does not support any particular scenario. There is a significant temporal mismatch between the ozone sounding and the lidar observations. The discussion is somewhat vague but may indicate that there is a significant change in the transport history during the time between the two observations. Rather than saying that this is difficult to explain, you could point to the trajectories in Figure 17 to argue that there was a strong change in air masses. However, this depends on how the trajectories were set up, which is not explained.*

The RH is excessive in terms of stratospheric intrusions (Trickl et al., 2014-2016). This is now added.

*Lines 435ff: Please show that in a figure.*

We think that this statement is clear enough! We need the re-analysis to make sure.

*Lines 445: Why was single photon counting abandoned, if it was the superior technique?*

It was abandoned because of it could no longer be started from the computer. Changed!

*Line 448: Again: temporal and spatial proximity*

Again: The spatial proximity did not change! Just the temporal proximity was regarded.

*Lines 462ff: This is the first time that quantitative data are provided for a time period under investigation. Quantitative results like this should be given for all time periods. Furthermore, I would suggest to use the IFU DIAL as reference in all calculations, which makes it easier to compare the numbers and the sign of these numbers.*

This is not true! In all three sections (2000, 2009, 2018) the averages and standard deviations are explicitly given (lines 362-363 and 415-416 in the reviewed version of the manuscript). I understand that it could make sense to present the IFU DIAL as the reference. We preferred IFU-stations because the station values are supposed to be more accurate and serve as the reference for the lidar. These differences are distributed almost without bias and much smaller than those of MOHp. Thus, they need not be compared with MOHp. What matters is MOHp sonde - IFU DIAL.

*Line 468: What is meant by "derivative formation"?*

The ozone density in determined from the slopes of the lidar signals. We now write "differentiating the backscatter signals".

*Line 484: How do you justify the potential bias of the lidar?*

This is tentative due to some deviations found. A re-evaluation of three to four profiles could, indeed, yield slight improvement with respect to MOHp. We do not want to discuss this further before the re-analysis is done.

*Lines 488f: How do you justify the quality of the ozone data, if there is a variable bias to the lidar? I don't see a problem that there is a bias between the two. The goal of this paper should be to quantify it and possibly to speculate on its cause.*

Not all lidar ozone profiles deviate from the sonde. Thus, an average does not make sense. We just estimate roughly 5 ppb as the maximum deviation. A statistical analysis can be made after the revision. Please, note that the focus of this paper is on 2018, representing 2012 to February 2019!

*Line 490: Is +/- 2 ppbv the limit of the bias (this is how this sentence reads), or rather its statistical uncertainty? I suspect it is the latter, and the manuscript should justify this uncertainty estimate.*

The offsets were determined by comparison with the lidar ozone profiles. We prepared figures for all comparisons which show good agreement of the shifted MOHp ozone also with the station data in most cases. This is obvious for the figures presented here. However, this sentence was removed here in order to avoid confusion. A statement on the potential use of the station data for recalibration of the MOHp sonde data as a whole is given below.

*Line 497: This bias estimate should have been justified in the earlier sections. The only quantitative values shown in line 464 gives different values. Furthermore, the way this statement reads is that there is a high likelihood that there is no bias between the systems since the uncertainty is larger than the actual value. I don't think this is intended. A more detailed description of the bias is needed.*

As stated above, the numbers for the statistical evaluations for the three years do exist. Here, we present the overall bias for all three years which is smaller than that for 2018.

*Figures: The manuscript uses a large number of figures, providing a bit too much detail, while leaving out some important points. I would suggest combining Figures 4, 5, 8, and 9 into one figure with 4 panels similar to Figure 2. Likewise, Figures 15, 16, 18, and 20 could be combined.*

This does not reduce the number of figures. In addition, the different figures belong to different sections and should be printed on the same pages. However, we decided to combine Figs. 11 to 13. This allows one to compare the differences for the different concentration ranges visually.

*Each of these panels should show only the uncorrected sonde and lidar profiles and separated, while sharing the same vertical axis, the true difference profile on the right of each panel. If there is a reasonable statistics, maybe all difference profiles can be summarized in a new Figure showing one single average difference. This may need to be done for each period or possible it can be done for the entire multi-decade period.*

As written above the true difference profile does not always make sense, particularly in summer. We prefer the way of presentation made since it allows to compare the profiles directly

*All Figures showing RH should have the axis label "Ozone [ppbv] / RH [%]". The RH should not be scaled in any of the Figures. Figures 5, 8, 18 and 19 use a scaling factor of 0.5; Figures 1 and 16 do not.*

We adopted these suggestions. We had preferred this style because the focus of the manuscript is on ozone and because we wanted to avoid crossing or co-located curves. However, the RH curves in the new figures look acceptable.

*I do not believe that showing the Raman lidar water vapor mixing ratio in Figure 2 adds much value. The sonde relative humidity might be more useful.*

We now show the CFH profiles of the $H_2O$ mixing ratio in addition to the lidar $H_2O$. This gives even better insight into the variability.

*Figure 2: I assume that the ECC ozone sonde profiles are not corrected. The absence of a significant bias could be pointed out in the text.*

Done!

*Figures 10, 14, and 20: It would be better to show "UFS-IFU" and, respectively, the other surface stations minus IFU. This would bring the MOHP-IFU in better context. The average lines are slightly misleading, in particular the smoothed "seasonal" bias in Figure 10. I would suggest removing these average lines from the Figures and discussing the averages quantitatively in the text.*

See above: For the DIAL (IFU) the stations are the reference, for MOHp the lidar! The average line was removed.

*Figures 11-13: These are misleading, since the bias has already been removed. It is important to see the scatter of the bias profiles to bring the vertical variability into the proper context. I suspect that spatiotemporal matching may be responsible for much of the vertical structure, i.e. it is of random nature and not related to any either instrument. If you feel that there is an instrumental component to the vertical structure of the bias, then you need to show and justify it.*

The quality of the sonde profiles is obviously high after removing the offset. We need this for a judgement of the lidar profiles. Thus, such a comparison is quite meaningful.

**Technical comments:**

*Line 23: "Side-by-side soundings"*

No necessarily! But changed!

*Line 28: Delete "These"*

We replace "These" by "Our"

*Line 62: Do you mean "sampling bias", when you mention "fair weather bias"?*

"sampling" added!

*Line 92: extra period*

????

*Lines 106f: Move coordinates directly following "MOHP (975 m a.s.l., 47.80 N, 11.00 E)".*

Moved!

*Line 192: Missing comma: "Since the 1990s,"*

We inserted a comma.

*Line 192: "TEI-49 ozone analyzers"*

"ozone" inserted.

*Line 194: "TEI-49iPS"*

I copied this paragraph from a manuscript of H. E. Scheel who passed away in 2013. The station was dismantled recently. Thus, I can no longer have a look.

*Lines 202f: Please rephrase. A TEI49C-PS instrument cannot be applied to something. In addition, it is not clear, whether this primary standard belongs to UFS and is used to recalibrate the TEI49i*

*weekly and monthly, or whether this instrument belongs to the UBA and is used to recalibrate the UFS ozone monitor annually.*

The instrument is a primary standard operated at UFS by UBA. We modified the text.

*Line 206ff: Better write: "The measurements were supported by a second instrument (Horiba APOA-370), which is equivalent to the TEI-49. GAW audits at the station for surface ozone took place in 2001, 2006 and 2011."*

Changed! We now also cite the reports of the audits.

*Lines 229f: Better write: "This indicates a spatial inhomogeneity of the air mass."*

Thank you! Changed!

*Line 231: Better write: "Different air masses must be assumed at different altitudes ..."*

Adopted!

*Line 235: Missing comma: "... were terminated, ..."*

Not in this case: The seonc part of the sentence does not make sense without the first one.

*Line 245: Delete "The first night of the campaign was clearer."*

Changed as suggested by Reviewer 1 ("clear" instead of "clearer").

*Lines 259ff: Better write: "In this altitude range the MR do not agree quantitatively with those obtained with the CFH sondes, because of the 1-h data-acquisition time necessary for the stratosphere."*

Changed!

*Line 261f: I would suggest to either delete this statement or elaborate. A discussion of the air-mass matching and a listing of how well the profiles match in time and space would be useful anyway.*

This explanation is given in the Introduction (lines 79-87 (81-89 in next manuscript)). Thus, no more words are needed.

*Line 295: probably need to change to "... spatiotemporal proximity ..."*

We added "spatial and".

*Line 298: Better write: "... This is explained by less structure in the ozone profile and ..."*

Changed!

*Line 361: Comma missing after "2018,"*

The second part of sentence does not make sense without the first one which prohibits a comma.

*Lines 466f: Better write: "Tropospheric differential-absorption ozone lidar systems are highly sensitive ..."*

The sentence about the bad reputation better introduces the issues.

*Lines 471f: Better write: "Based on continual improvements, starting with the 1994 system upgrading, the IFU ozone DIAL reached peak performance by 2012, but potential for minor improvements remains. Comparison ..."*

Good suggestion, changed!

*Line 476: Better write: "Here, we analyse the lidar performance in three periods during its technical development."*

Changed, including the suggestion of Reviewer 1.

*Line 479: Delete "perfect"*

Deleted!

*Line 483: Better write: "Between 2007 and 2011, we suspect a slight negative …"*

Changed!

*Line 486: Delete "just"*

Deleted!

*Line 488: replace "identify" with "validate"*

Replaced!

**Review 3**

*The paper of Trickl et al. is of interest to a broad community of researchers interested in natural variability in free tropospheric ozone as well as the development of reliable tropospheric ozone instrumentation. The analysis is very good. The only concern at initial review is that the manuscript does not conform to FAIR principles (findable, accessible, interoperable, reusable). Thus, publication is recommended when all data are openly archived and meet the FAIR criteria.*

As mentioned, the MOHp data are publicly available on the NDACC server. The FZJ data are also publicly available. We forgot to include the summit data in this section that are available on two data bases. This is now done. Just the lidar data are only available on request at this time since (as mentioned) a partial revision is required as a result of this study. In general: IFU is not a public environmental office just producing data. As a research institute it is reviewed based on the scientific output. We have repeatedly had bad experience with groups using our freely available data without offering a co-authorship or at least citing our publications.

---

## Referee Report (RR1)

**Review of AMTD-manuscript entitled "Local comparisons of tropospheric ozone: Vertical soundings at two neighbouring stations in Southern Bavaria" by Thomas Trickl et al.**

General remarks:

The content of the manuscript has already been well described by the previous three referees who had reviewed the first version of the manuscript, such that I will constrain my review mostly to the major and minor revisions demanded by referee #2. Regarding the minor revisions, the authors have responded and revised the manuscript appropriately, however, the new manuscript still lacks in the way the comparisons between lidar and ozonesondes were made. The major critics of referee#2 was that in multiple figures and descriptions in the text bias corrections to the ozonesonde data were applied without given the reason/cause for that. Therefore, it would be more interesting to properly describe the bias and show profiles of the actual differences. I fully share these demands of referee#2. Unfortunately, in the revised manuscript the authors did not revise the text and figures adequately, neither replied in a satisfying way to these major critics made by referee#2, which I fully underline. In the revised manuscript the authors still follow their original methodology to do unexplained bias corrections to the ozone data. When doing such corrections then solid arguments have to be given, which are still missing. At present, the reader easily gets the impression that the bias corrections are just artificial "corrections" to adjust the comparisons to get a better agreement of the Lidar with the sondes, however, this would finally not improve the trust in the Lidar data, but just do the opposite. The last is certainly not in the authors their own interests, and therefore I strongly recommend to going for a second revision of the manuscript, but now follow the major comments given by referee#2 more strictly.

---

## Author Response (AR2)

**Reply to Second Review of "Local comparisons of tropospheric ozone: Vertical soundings at two neighbouring stations in Southern Bavaria" by Trickl, et al.**

ThomasTrickl

September 14, 2023

**General remarks**

Given the substantial changes made during the revision I am astonished to read the lines of Reviewer 2. Nevertheless, I tried to make amendments in order to improve the clearness.

One again, I print the lines of the reports in italics and my replies in normal text.

**Report 1 (July 31, 2023):**

Some minor corrections

1) *line 30. Correct to "...for low to moderate ozone concentrations"*

Corrected.

2) *line 543: '... differentiating the backscatter signals.". Please cite a relevant publication.*

I found this phrase in line 515: I cited our 2020 lidar paper.

3) *line 575: "... Scheel for 3 km". please provide a relevant citation.*

I already wrote "see Introduction", which refers to the statement "personal communication around 2010".

**Report 2 (July 20, 2013):**

*Summary:*

*The revised version of the manuscript accepted many of suggestions for improvement. Thank you. However, my main major comment was largely ignored and therefore, the paper remains slightly confusing and in some places misleading. I do object to ==adjusting ozone== sonde profiles to lidar data and then using these adjusted profiles to evaluate the quality of the lidar. By adjusting the ozone profiles to the lidar, there is already the implicit assumption ==that the lidar is a suitable reference==. Once this assumption has been made, the result cannot be used to prove that the lidar is a suitable reference.*

I am astonished to read that we adjust the MOHp sonde data to the lidar data implicitly assuming that the lidar is a suitable reference. Indeed, the lidar is a suitable reference as demonstrated in Sec. 3.1! We made clear statements about this, e.g. in lines 314-316 in the reviewed version. Nevertheless, thank you for pointing this out! I tried to clarify this more explicitly.

In the "Results" section we first describe the highly successful comparison with the FZJ ECC sondes that, together with the UFS data prove the excellent performance of the lidar. The agreement with the mountain stations has been routinely verified over many years (as mentioned several times, starting with the Introduction), which is confirmed and statistically evaluated in the current study. There is an indication that the lidar calibration could be free of bias. The unique side-by-side comparison of lidar and ECC sondes plus the station data yield just minor concentration offsets of these sondes, but an impressive agreement in vertical structure.

Based on this successful validation of the lidar we start the comparison with the MOHp sonde. Despite the distance of 38 km between both sites we find structural agreement between the soundings, but also offsets that change from sonde to sonde. It is hard to believe that these offsets are caused by accidentally vertically agreeing atmospheric differences. This is now discussed in the introduction to Sec. 3.2. We evaluate both the agreement of the lidar with the mountain sites and the offsets statistically. Since the agreement of lidar and in-situ ozone is convincing we conclude that the offsets are a sonde-specific issue. Only after this conclusion we try to evaluate an upper limit of the lidar uncertainty based on the retrieved structural differences.

*This does not mean that I consider any of the instruments of deficient quality. Each of the measurements systems has their strengths and weaknesses. Evaluating the quality of each means describing their differences (biases or offsets) relative to each other (without adjusting one or the other). ==The attribution of the offsets should be done based on additional information, for example the surface sites, but other information may be suitable as well.==*

I am really astonished to read this sentence since this is what was actually done (e.g., lines 387-396 of the old manuscript)! As mentioned above, I added some information including the routine comparisons with the summit sites and UFS over many years of ozone sounding with the DIAL. Many examples have been shown in our earlier publications. In this manuscript we even present a statistical analysis for all the three years of comparison.

*The revision still does not clarify the ==hierarchy of references==. This is a source of the confusion. While the authors may have a clear picture, which instruments they consider the most reliable as reference for which purpose, this does get lost in the order data are being presented and discussed.*

As pointed out above I do not agree. However, although this hierarchy is clearly visible I add more explicit statements.

*I consider this manuscript an important paper for the ozone monitoring community. However, the analysis and presentation of the material confuses issues more than it helps elucidate the current state of observing capabilities. I would strongly urge the authors to take another look. In its current stage, the manuscript still requires major revisions.*

*Detailed comments:*

*Lines 283 ff: I am not sure, whether you have understood my comment during the initial review: It is not appropriate to adjust data and then call an agreement outstanding. Either data have to be adjusted to reach agreement (i.e. there is no agreement), or the agreement (without adjustment) is outstanding. You cannot have it both ways!*

It is very difficult indeed to follow this argument. The outstanding agreement is not claimed for the unshifted data. I revised that part.

*ECC ozone sondes are generally considered absolute instruments, i.e. they do not require calibration (against a known ozone reference). This implies that the term "uncalibrated" is not applicable here. Rather than showing an agreement that was forced, the magnitude and the character of the differences need to be described. This may be important information for the ECC community.*

Thank you for this remark. As can be found in Sec. 2.3 the ECC sondes were prepared as described and (as I learnt) no ground calibration was done. This is what "uncalibrated" means. However, the entire paragraph has changed and "uncalibrated" no longer exists.

*Lines 307 ff: Same as comment on Lines 283ff. In addition, which instrument is evaluated against which. The start of this paragraph reads: "For quantifying the quality of the lidar measurements*

*...". Next you proceed to correct the ECC ozone sonde profiles based on the lidar comparison. Then, they describe the small offsets between the two systems. Lastly, in that paragraph they conclude: "This result justifies to use the lidar as a quality standard in the comparisons with the MOHp Brewer-Mast sondes described in the following sections." ==It is not appropriate to adjust one instrument to another and then use the resulting agreement as quality justification in the comparison with a third instrument.== I do agree that the lidar is a good instrument. I disagree with the logic of the argument.*

Thank you! I tried to describe the hierarchy clearer than before. The lidar agrees better with UFS than the ECC sondes. However, I think that offsets of 1.5 ppb ± 1 ppb are not a bad result for the ECC sondes. As to the comparisons with MOHp I see reasonable agreement after removing the offsets. It is an important question why this approach looks rather suitable. Given the amount of material analysed I cannot believe in accidentally constant atmospheric offsets. Of course an altitude-dependent atmospheric component is present and cannot be removed. As mentioned above, this is now more explicitly discussed.

*Lines 337f: "After adding 5.8 ppb the sonde results (cyan curve) match the lidar values well for altitudes above 2.1 km." This is the same issues as above, i.e. that you cannot have it both ways. However, here it is easier to correct. You could simply write: "The Brewer Mast ozone sondes show a low bias of 5.8 ppbv relative to the lidar above 2.1 km."*

Accepted.

*Lines 354ff: In the discussion of the summer profiles, this is same issue not as easily corrected. Most importantly, it is no longer possible to evaluate the uncertainty of the lidar measurements (lines 349f).*

This is obvious because of the atmospheric component. Thus, the analysis can give just an upper limit.

*Lines 428: Same issue as above and something I pointed out in the initial review. To be clear what I mean: The original ozone sonde data show a bias relative to the lidar. You remove this offset from the sonde data. And then you state that "The analyses for 2018 do not reveal a significant bias ...". Quite to the contrary, your analysis did find a bias and you removed it. Therefore, your statement is misleading. Unfortunately, the source of this confusion is your basic approach to remove the bias from the sonde measurements rather than just sticking to describing it.*

What means "describing"? We find that the approach of a constant bias rather suitable since it is verified in so many examples. Of course, the explanation of the bias is an interesting topic for future research. In any case, I modified this sentence.

*Figures 11 (formerly Figs 11-13): This Figure relates to the fundamental presentation challenge that I tried to raise in my initial review and above. It is difficult to evaluate differences, if they have been removed in an earlier stage. What do the remaining differences tell the reader? This is unfortunately misleading and has not been addressed.*

It is reasonable to assume that the Brewer-Mast sonde reproduces the vertical structure of tropospheric ozone, apart from the bias. This is verified in most of the comparisons, although local differences exist at some altitudes. We clearly point out that the uncertainties are overestimated due to the atmospheric issues.

*Line 33: You did not address my initial question. What does the uncertainty, the value after the +-refer to? You now use the term "maximum of deviations". In the text, you actually specify the*

*standard deviation without specifying the "maximum of deviations". It would help the manuscript to make these two identical.*

We now take the standard deviation throughout the Abstract since this is the quantity derived in the analyses.

*Line 134: Your answer to my initial comment is interesting. Please include this brief discussion in the manuscript at the appropriate location.*

Information on sonde comparisons is given in the Introduction. One sentence was added to the Discussion.

*Lines 446ff: I don't believe you understood what I meant in my initial comment. Although the distance between the stations is fixed, ozone gradients in the atmosphere are not. If winds are blowing along the line connecting both stations, the spatial separation is probably negligible. If winds are blowing orthogonal to that line, then the spatial separation is quite important, despite their relative proximity.*

I fully agree. We discuss here just two examples with substantial discrepancies. However, I have problems in assuming that an atmospheric structure extends over the entire troposphere to explain the constant bias. I added a sentence on this.

*Technical comments:*

*Line 26: Delete "just"*

Removed

*Line 133: Change to "(corresponding to more than 2.5 ppb)"*

Changed.

*Line 175: Delete "just"*

Deleted.

*Line 263f: Change to "(indicated by low relative humidity)"*

Changed.

*Line 266: Delete "must be assumed"*

Deleted.

*Line 425: "severest" -> "most severe"*

Changed!

**Report 3 (July 31, 2023)**

*General remarks:*

*The content of the manuscript has already been well described by the previous three referees who had reviewed the first version of the manuscript, such that I will constrain my review mostly to the major and minor revisions demanded by referee #2. Regarding the minor revisions, the authors have responded and revised the manuscript appropriately, however, the new manuscript still lacks in the way the comparisons between lidar and ozonesondes were made. The major critics of referee#2 was that in multiple figures and descriptions in the text bias corrections to the ozonesonde data were applied without given the reason/cause for that. Therefore, it would be more interesting to properly describe the bias and show profiles of the actual differences. I fully share these demands of referee#2. Unfortunately, in the revised manuscript the authors did not revise the*

*text and figures adequately, neither replied in a satisfying way to these major critics made by referee#2, which I fully underline. In the revised manuscript the authors still follow their original methodology to do unexplained bias corrections to the ozone data. When doing such corrections then solid arguments have to be given, which are still missing. At present, the reader easily gets the impression that the bias corrections are just artificial "corrections" to adjust the comparisons to get a better agreement of the Lidar with the sondes, however, this would finally not improve the trust in the Lidar data, but just do the opposite. The last is certainly not in the authors their own interests, and therefore I strongly recommend to going for a second revision of the manuscript, but now follow the major comments given by referee#2 more strictly.*

In tried to describe the hierarchy of the instruments in more detail.

The offset corrections bring both instruments in substantially better agreement. Since the lidar agrees well with the ECC sonde and the station data we can assume that the lidar produces highly accurate results, at least under conditions up to moderate ozone. The agreement with the station data persists throughout the period under investigation. Thus, in this altitude range we are sure about what we are doing. Our concern has been the range above 6 km, particularly in summer. Here, our results suggest a reasonable agreement, apart from some issues in 2009.

Of course, we do not ultimately know if there are systematic differences in air-mass composition between the lidar an MOHp. However, since the agreement of the profiles in structure is so good on average it is hard to believe that such a difference exists. We cannot fully remove the atmospheric variability. Thus, out statistics yield just an upper limit of the uncertainty of the lidar.